# Microtubules restrict F-actin polymerization to the immune synapse via GEF-H1 to maintain polarity in lymphocytes

Judith Pineau[1,2], Léa Pinon[1,3,4], Olivier Mesdjian[3,4], Jacques Fattaccioli[3,4], Ana-Maria Lennon Duménil[1]*, Paolo Pierobon[1]*

[1]Institut Curie, PSL Research University, INSERM U932, Paris, France; [2]Université de Paris, Paris, France; [3]Laboratoire P.A.S.T.E.U.R., Département de Chimie, École Normale Supérieure, PSL Research University, Sorbonne Université, CNRS, Paris, France; [4]Institut Pierre-Gilles de Gennes pour la Microfluidique, Paris, France

**Abstract** Immune synapse formation is a key step for lymphocyte activation. In B lymphocytes, the immune synapse controls the production of high-affinity antibodies, thereby defining the efficiency of humoral immune responses. While the key roles played by both the actin and microtubule cytoskeletons in the formation and function of the immune synapse have become increasingly clear, how the different events involved in synapse formation are coordinated in space and time by actin–microtubule interactions is not understood. Using a microfluidic pairing device, we studied with unprecedented resolution the dynamics of the various events leading to immune synapse formation and maintenance in murine B cells. Our results identify two groups of events, local and global, dominated by actin and microtubules dynamics, respectively. They further highlight an unexpected role for microtubules and the GEF-H1-RhoA axis in restricting F-actin polymerization at the lymphocyte–antigen contact site, thereby allowing the formation and maintenance of a unique competent immune synapse.

*For correspondence:
Ana-Maria.Lennon@curie.fr (A-MLD);
paolo.pierobon@curie.fr (PP)

**Competing interest:** The authors declare that no competing interests exist.

## Editor's evaluation

This study provides new insights into the fundamental process of immune cell synapse formation in the context of B-lymphocyte antigen receptors and cognate antigen-presenting cells or surfaces. The authors use elegantly designed microfluidic systems, allowing control of the nature and number of the antigenic surfaces presented, and at the same time an unprecedented view of the process of immune synapse maturation in situ. These experiments provide an understanding of the specific roles of the reorganization of the actin as well as the microtubule cytoskeleton in the selection and restriction of a unique immune synapse, an important process in high-affinity antibody generation.

## Introduction

Cell polarization refers to the acquisition of a cell state characterized by the asymmetric distribution of cellular individual components, including molecules and organelles. It is critical for a multitude of cellular functions in distinct cell types and further controls cell–cell interactions. This particularly applies to lymphocytes, which rely on cell polarity to form a stereotyped structure called the immune synapse to communicate with antigen-presenting cells (*Monks et al., 1998*; *Dustin et al., 1996*; *Fleire et al., 2006*; *Carrasco and Batista, 2007*; *Junt et al., 2007*). Immune synapses are not only

instrumental for lymphocyte activation but also serve their effector functions, for example, by facilitating the killing of infected or malignant cells by cytotoxic cells (*Potter et al., 2001*; *Batista and Dustin, 2013*). Understanding how immune synapses form has thus become a major challenge for cell biologists and immunologists for the last decade, yet many mechanistic questions remain unanswered. In particular, how immune synapses are maintained in time to serve sustained lymphocyte function and allow robust immune activation is poorly understood.

Immune synapse formation is accompanied by the reorganization of lymphocyte antigenic receptors and associated signaling molecules into a concentric structure that forms at the contact zone with antigen-presenting cells (*Monks et al., 1998*; *Fleire et al., 2006*). The synapse allows the exchange of information (molecules and vesicles) between the two cells through tightly regulated exocytic and endocytic events (*Griffiths et al., 2010*). Signaling and trafficking at the immune synapse require deep rearrangements of both the lymphocyte actin and microtubule cytoskeletons (*Douanne and Griffiths, 2021*). On one side, the actin cytoskeleton controls the organization of antigen receptor-containing microclusters for coordination between trafficking and signaling and further helps generating the mechanical forces that depend on the myosin II motor (*Treanor et al., 2010*; *Treanor et al., 2011*; *Kumari et al., 2019*; *Bolger-Munro et al., 2019*). On the other side, the microtubule cytoskeleton controls the recruitment of organelles at the immune synapse. This relies on centrosome reorientation, leading to lymphocyte symmetry breaking and acquisition of a polarized cell state (*Yuseff et al., 2011*; *Torralba et al., 2019*). Although it is now clear that these events of actin and microtubule reorganization are instrumental for synapse formation, how they depend on each other and are coordinated to ensure proper and durable synapse function remains elusive.

There is growing evidence in the literature suggesting that the actin and microtubule cytoskeletons do not act independently of each other but indeed functionally and/or physically interact (*Dogterom and Koenderink, 2019*; *Hohmann and Dehghani, 2019*). This is well-illustrated, for example, by the study of oocyte polarization in *Caenorhabditis elegans* where polarization of intracellular organelles occurs in response to actomyosin contraction at one cell pole, which is in turn downregulated upon centrosome recruitment (*Gubieda et al., 2020*). A crosstalk between actin and microtubules in lymphocytes was also recently highlighted by our work, showing that clearance of branched actin at the centrosome is needed for its detachment from the nucleus and polarization to the synapse (*Obino et al., 2016*). However, whether the microtubule network in turn impacts on actin dynamics and immune synapse formation, function, and maintenance has not been studied, in part because the tools to quantitatively monitor in time both local actin reorganization and microtubule reorientation were not available so far. In this work, we developed a microfluidic chamber to quantitatively analyze both the local and global events associated to immune synapse formation in time and space and establish their dependency on actin and microtubule cytoskeletons. Our results revealed that the microtubule network controls the polarized polymerization of F-actin at the interface between lymphocytes and antigen-presenting cells, thereby allowing sustained formation of a unique and functional immune synapse.

## Results

### A microfluidic system for the systematic study of immune synapse formation

We aimed at understanding how local and global events of synapse formation were coordinated in space and time. As a model, we used B lymphocytes, which form immune synapses upon engagement of their surface B cell receptor (BCR) by cognate antigens presented at the surface of neighboring cells. In vivo, this cell–cell interaction takes place in lymphoid organs and is required for antigen extraction and activation of signaling pathways that later on promote B lymphocyte differentiation into cells able to produce high-affinity antibodies (*Carrasco and Batista, 2007*; *Pape et al., 2007*). Antigen extraction involves two modes: (1) an early mechanical mode that relies on actin-mediated forces at the synapse and (2) a late proteolytic mode that requires centrosome polarization to the synapse and subsequent lysosomes transport on microtubules and secretion of hydrolases into the extracellular milieu (*Yuseff et al., 2011*; *Natkanski et al., 2013*; *Spillane and Tolar, 2016*). It has been shown that mechanical antigen extraction occurs on deformable substrates while proteolytic extraction is used to extract antigen from stiff materials (*Spillane and Tolar, 2016*). The first pathway, when activated,

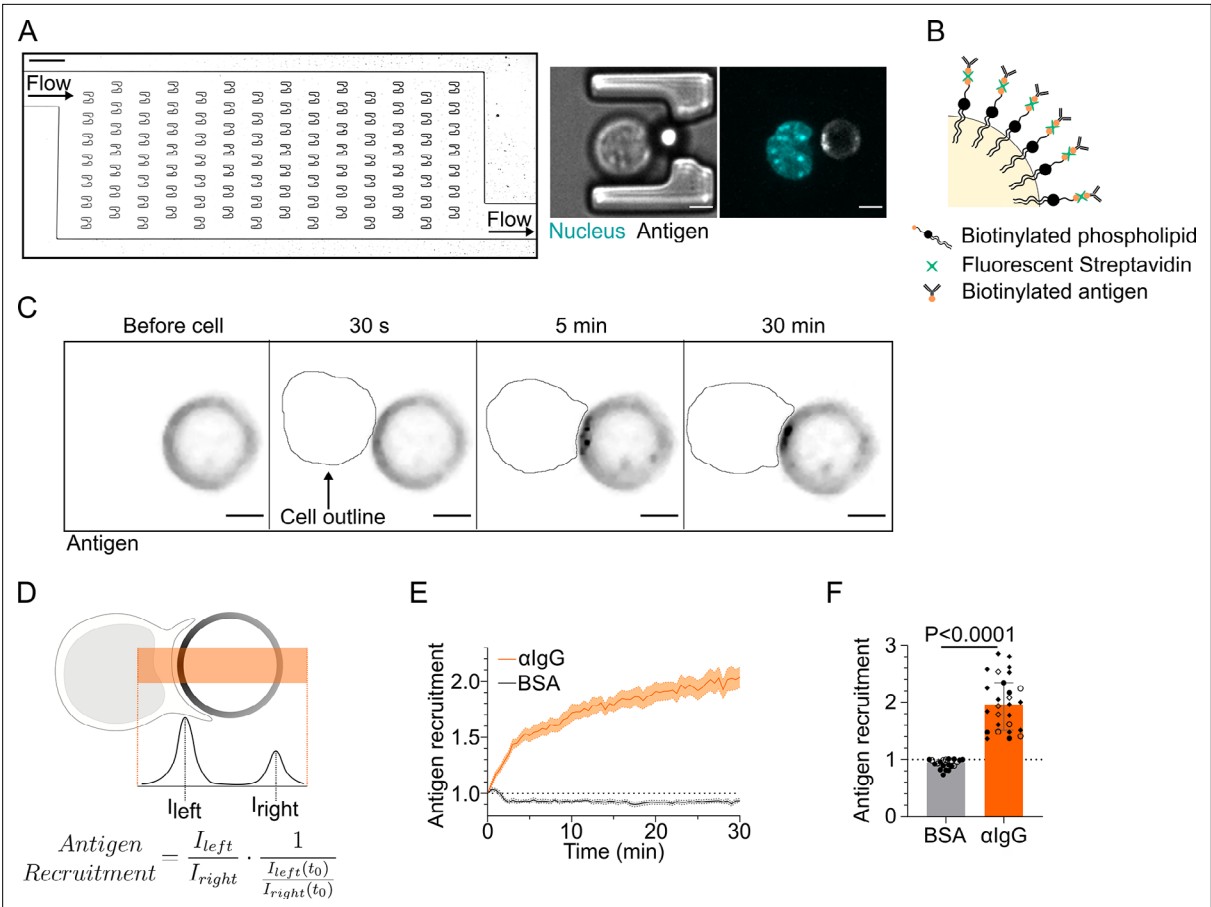

**Figure 1.** Microfluidic system to study dynamics of B lymphocyte polarization and immune synapse formation. (**A**) Transmission image of a chamber of the microfluidic chip containing the traps. Scale bar 100 μm. Inset: cell–droplet doublet in a microfluidic trap. Bright-field image and fluorescence image (nucleus: cyan; antigen: gray). Scale bar 5 μm. (**B**) Schematic representation of the surface of an oil droplet used for antigen presentation. (**C**) Time-lapse images of antigen recruitment on a F(ab')₂αIgG-coated droplet (acting as an antigen). Scale bar 5 μm. (**D**) Schematic representation of the quantification of antigen recruitment at the immune synapse. (**E**) Quantification over time of recruitment on BSA-coated (negative control) or αIgG-coated droplets at the immune synapse (median ± IQR) and (**F**) plateau of antigen recruitment (average value 25–30 min) on BSA- or αIgG-coated droplets (mean ± SEM, BSA N = 14;7, αIgG N = 4;15;4,4, pooled from >2 independent experiments, Mann–Whitney test).

The online version of this article includes the following source data and figure supplement(s) for figure 1:

**Figure supplement 1.** Microfluidic traps and antigen-coated droplets allow the study of the B cell immune synapse in cell lines and primary B cells.

**Source data 1.** Data tables related to graphs in *Figure 1*.

**Figure supplement 1—source data 1.** Data tables related to graphs in *Figure 1—figure supplement 1*.

inhibits the second one (*Spillane and Tolar, 2016*), suggesting a functional interaction between these actin- and microtubule-dependent events. However, the experimental systems used so far did not allow to reach a sufficient temporal resolution to quantitatively monitor the evolution of both cytoskeleton networks in 3D from the first instant of immune synapse formation. To circumvent this problem, we built a microfluidics device based on an array of traps where antigen-coated oil droplets and B cells can be sequentially captured (*Figure 1A*, *Video 1*). Antigen-coated lipid droplets are a good 3D substrate to mimic antigen-loaded cells as they allow antigen mobility at their surface (*Figure 1B*). Moreover, they are effectively stiff (see 'Materials and methods') and might thus also allow lysosome recruitment at the synapse and proteolytic antigen extraction. Chambers were imaged in 3D from the time of cell injection to capture the entire process of synapse formation. Droplets were functionalized either with a non-activating molecule (BSA, negative control) or an activating BCR ligand (F(ab')₂ anti-mouse IgG, referred to as 'antigen' from now on). Both ligands were grafted to the lipid droplet with fluorescent streptavidin to follow their accumulation dynamics at the droplet surface (*Figure 1B–D*, *Video 2*). Such an accumulation was exclusively observed upon engagement of the BCR with its

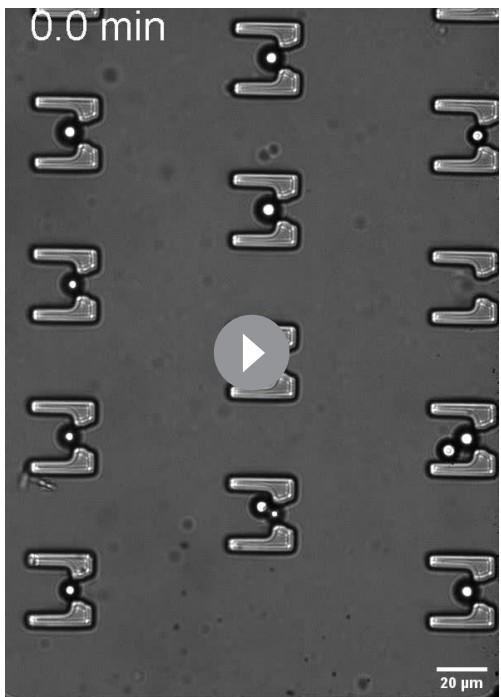

**Video 1.** Bright-field movie of cell injection in the microfluidic chip.

https://elifesciences.org/articles/78330/figures#video1

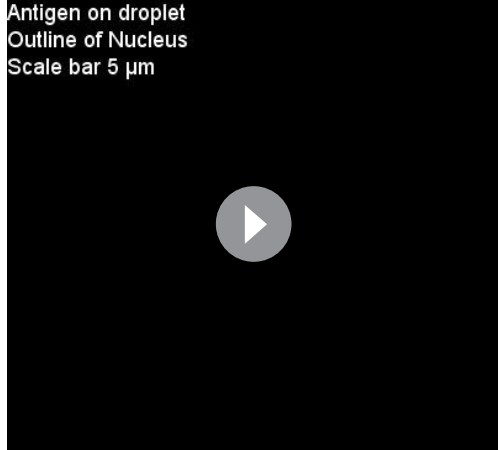

**Video 2.** Recruitment of antigen on the droplet by a IIA1.6 cell; outline of the nucleus drawn to follow cell arrival.

https://elifesciences.org/articles/78330/figures#video2

ligand, BSA-coated droplets remaining homogeneously fluorescent (*Figure 1E and F*). Staining of the exocyst component EXOC7 implicated in lysosomal proteases secretion at the synapse (*Yuseff et al., 2011*; *Sáez et al., 2019*) showed an enrichment of this protein 45 min upon activation (*Figure 1—figure supplement 1A*), suggesting synapse functionality in terms of antigen extraction. Of note, we confirmed that both antigen and actin were enriched at the immune synapse of primary murine IgM⁺ B cells in the first minutes after BCR engagement (*Figure 1—figure supplement 1B–E*), showing that these observations are not restricted to our model B cell line. Altogether, these results indicate that our microfluidics system can be used to study the dynamics of immune synapse formation as well as the mechanisms involved in its maintenance.

## Defining characteristic timescales of immune synapse formation

Our microfluidic system was used at first to visualize and extract the typical timescales of the key events associated to synapse establishment: BCR signaling (production of diacylglycerol [DAG] monitored by a GFP-C1δ reporter; *Botelho et al., 2000*), F-actin reorganization (labeled with F-tractin-tdTomato), centrosome (labeled with SiRTubulin), and Golgi apparatus (labeled with Rab6-mCherry) polarization, lysosomes (labeled with LysoTracker), and nucleus (labeled with Hoechst) repositioning. Characteristic timescales were extracted from volumetric images taken every 30 s (*Video 3*). We found that the peak of DAG production occurred ~3.25 min upon contact between the lymphocyte and the antigen-coated droplet (*Figure 2A and G*, *Figure 2—figure supplement 1*). This time is comparable to the one found in *Gawden-Bone et al., 2018* for cytotoxic T cells. This event was concomitant with actin polymerization, which peaked at the synapse at ~3 min (*Figure 2B and*

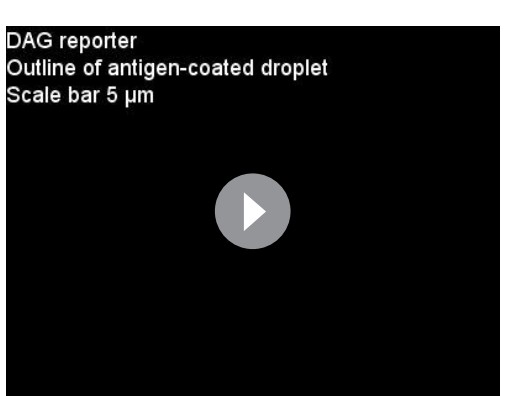

**Video 3.** Examples of polarization dynamics at the B cell immune synapse of a IIA1.6 cell, for diacylglycerol (DAG) signaling, F-actin, the centrosome, the Golgi apparatus, lysosomes and the nucleus; droplet outline drawn on each movie.

https://elifesciences.org/articles/78330/figures#video3

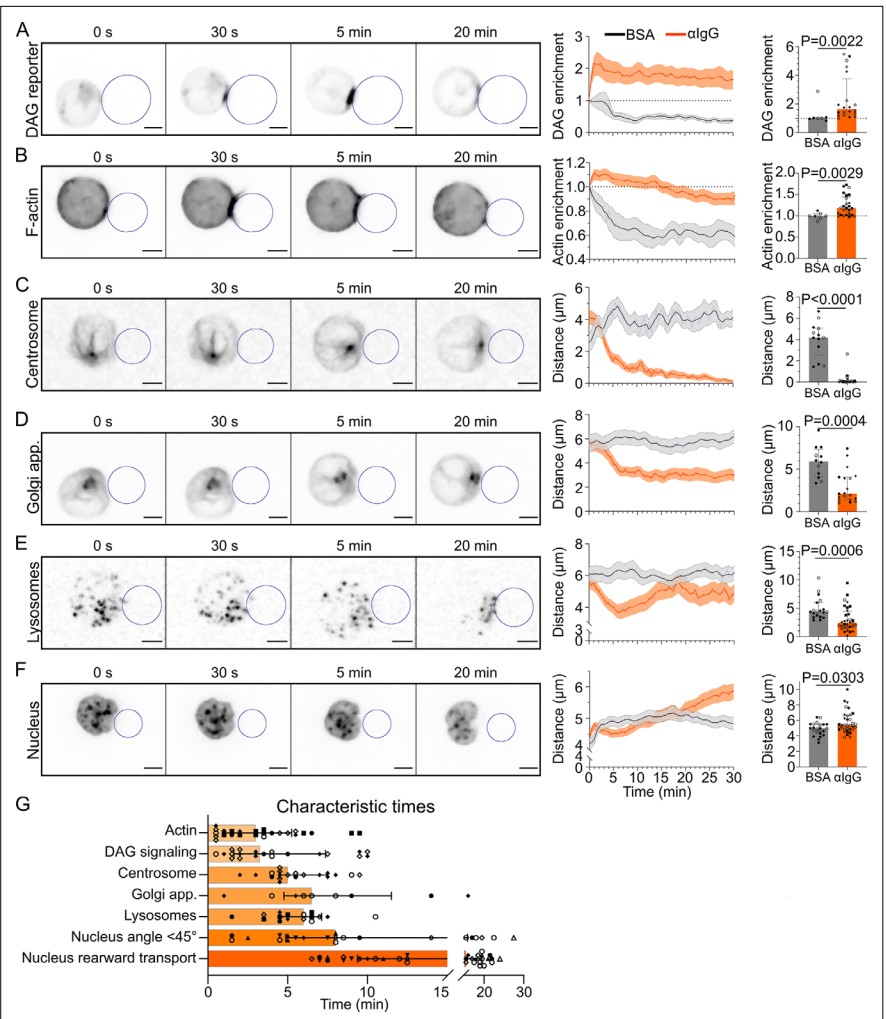

**Figure 2.** Timescales of B lymphocyte polarization. All images in this figure are from 3D SDCM time-lapse imaging of IIA1.6 cells in contact with an antigen-coated droplet (outlined in blue). Analyses were done in 3D. (**A**) Time-lapse images of a IIA1.6 cell expressing a diacylglycerol (DAG) reporter (C1δ-GFP), in contact with an antigen-coated droplet. Enrichment in time of DAG reporter, defined as the intensity within 1 μm of the droplet, normalized by this value at $t_0$ (mean ± SEM). Maximum enrichment (0–10 min) (median ± IQR, pooled from >2 independent experiments, BSA: N = 4;3, αIgG: N = 2;2;7;9, Mann–Whitney test). (**B**) Time-lapse images of a IIA1.6 cell expressing F-tractin-tdTomato, in contact with an antigen-coated droplet. Enrichment in time of F-actin defined as the intensity within 2 μm of the droplet divided by the intensity in the whole cell, and normalized by this value at $t_0$, for BSA- or αIgG-coated droplets (mean ± SEM). Maximum enrichment (0–10 min) (median ± IQR, pooled from >2 independent experiments, BSA: N = 2;5, αIgG: N = 4;2;3;6;10, Mann–Whitney test). (**C**) Time-lapse images of a IIA1.6 cell stained with SiRTubulin to visualize the centrosome, in contact with an antigen-coated droplet. Distance over time between the centrosome and droplet surface for BSA- or αIgG-coated droplets (mean ± SEM). Average plateau distance (25–30 min) (median ± IQR, pooled from >2 independent experiments, BSA: N = 8;5, αIgG: N = 2;3;12;8, Mann–Whitney test). (**D**) Time-lapse images of a IIA1.6 cell expressing Rab6-mCherry to visualize the Golgi apparatus, in contact with an antigen-coated droplet. Distance over time between the Golgi body and droplet surface for BSA- or αIgG-coated droplets (mean ± SEM). Average plateau distance (25–30 min) (median ± IQR, pooled from >2 independent experiments, BSA: N = 9;3, αIgG: N = 4;1;8;6, Mann–Whitney test). (**E**) Time-lapse images of a IIA1.6 cell stained with LysoTracker to visualize lysosomes, in contact with an antigen-coated droplet. Average distance over time between lysosomes and droplet surface for BSA- or αIgG-coated droplets (mean ± SEM). Minimum distance (3–10 min) (median ± IQR, pooled from >2 independent experiments, BSA: N = 13;6, αIgG: N = 3;5;10;5;9, Mann–Whitney test). (**F**) Time-lapse images of a IIA1.6 cell stained with Hoechst to visualize the nucleus, in contact with an antigen-coated droplet. Nucleus–droplet distance in time (mean ± SEM). Average distance in the final state (25–30 min) (median ± IQR, pooled from >2 independent experiments, BSA: N = 14;9, αIgG: N = 5;10;2;7;5;1;4, Mann–Whitney test). (**G**) Characteristic times of polarization events, extracted from

*Figure 2 continued on next page*

*Figure 2 continued*

the data of (**A–F**) and *Figure 3*. $N_{DAG}$ = 2;2;7;9, $N_{Actin}$ = 4;2;3;6;10, $N_{Centrosome}$ = 2;2;8;5, $N_{Golgi}$ = 2;4;3, $N_{Lyso}$ = 2;3;3;4;6, $N_{Nuc\ angle}$ = 3;7;1;3;4;1;3, $N_{Nuc\ transport}$ = 5;10;2;7;5;1;4. Scale bar 5 µm.

The online version of this article includes the following source data and figure supplement(s) for figure 2:

**Figure supplement 1.** Single-cell kinetics of markers of B lymphocyte polarization.

**Source data 1.** Data tables related to graphs in *Figure 2*.

---

*G*, *Figure 2—figure supplement 1*). Formation of the stereotypical actin pattern, with actin protrusions at the periphery and an actin-cleared area at the center, was then observed. Centrosome and Golgi tracking over time showed that they displayed similar behaviors, reaching the immune synapse area after 5 min for the centrosome (distance <2 µm) and 6.5 min for the Golgi apparatus (distance <4 µm) (*Figure 2C, D and G*, *Figure 2—figure supplement 1*). This was only observed in cells where the BCR was specifically engaged and is in good agreement with these two organelles being physically associated (*Chabin-Brion et al., 2001*). Lysosomes, which are also known to associate with microtubules for intracellular transport, displayed a slightly different behavior: their distance to the immune synapse decreased down to ~3 µm in ~6 min, indicating their polarization, but then increased (*Figure 2E and G*, *Figure 2—figure supplement 1*), maybe due to the secretion of lysosomal vesicles, which would lead to signal fainting at the immune synapse and a consequential apparent redistribution all over the cell. Finally, we observed that the nucleus was transported to the rear of the cell at later timepoints (*Figure 2F*). Closer observation revealed that this organelle displayed a biphasic movement: a rotation reoriented the nucleus until its stereotypical lymphocyte nuclear invagination faces the immune synapse ($\theta_N$ < 45° after ~8 min); once the nucleus had reoriented, it started moving toward the cell rear ~15 min after contact with the droplet, slowly reaching the opposite cell pole over time (*Figure 2G*, *Figure 3A–D*). In summary, quantification from single kinetics of the various events leading to immune synapse formation in B lymphocytes suggests the existence of two groups

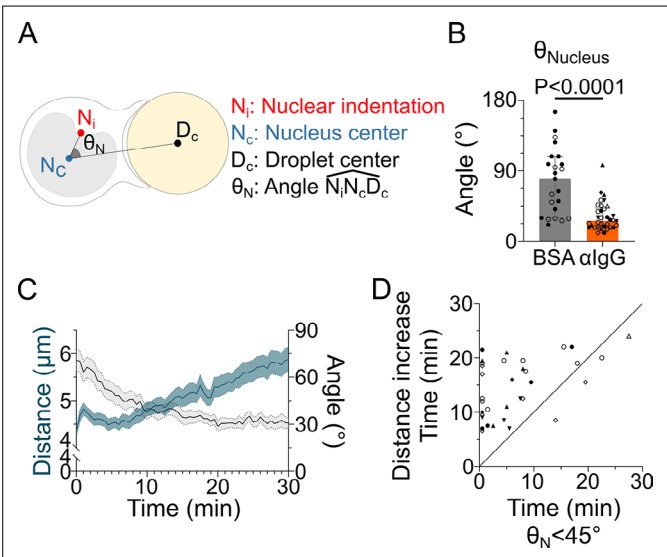

**Figure 3.** The nucleus undergoes a rotation followed by rearward transport. Analyses were performed on movies obtained from 3D SDCM time-lapse imaging of IIA1.6 cells stained with Hoechst, in contact with a F(ab')₂αIgG- or BSA-coated droplet. (**A**) Schematic defining the angle measured to assess nucleus orientation (analysis was done in 3D). The indentation was detected based on local curvature. (**B**) Average angle $\theta_N$ in the final state (25–30 min) (pooled from >2 independent experiments, median ± IQR, BSA N = 14;9, αIgG N = 5;10;2;7;5;1;4, Mann–Whitney test). (**C**) Overlay of nucleus–droplet distance and $\theta_N$ over time for cells in contact with αIgG-coated droplets and (**D**) time for which the cell reaches $\theta_N$ < 45° (invagination oriented toward the immune synapse), and time of last local minima of nucleus–droplet distance (time after which the nucleus is only transported to the rear) (same data as in **B**). Line at Y = X.

The online version of this article includes the following source data for figure 3:

**Source data 1.** Data tables related to graphs in *Figure 3*.

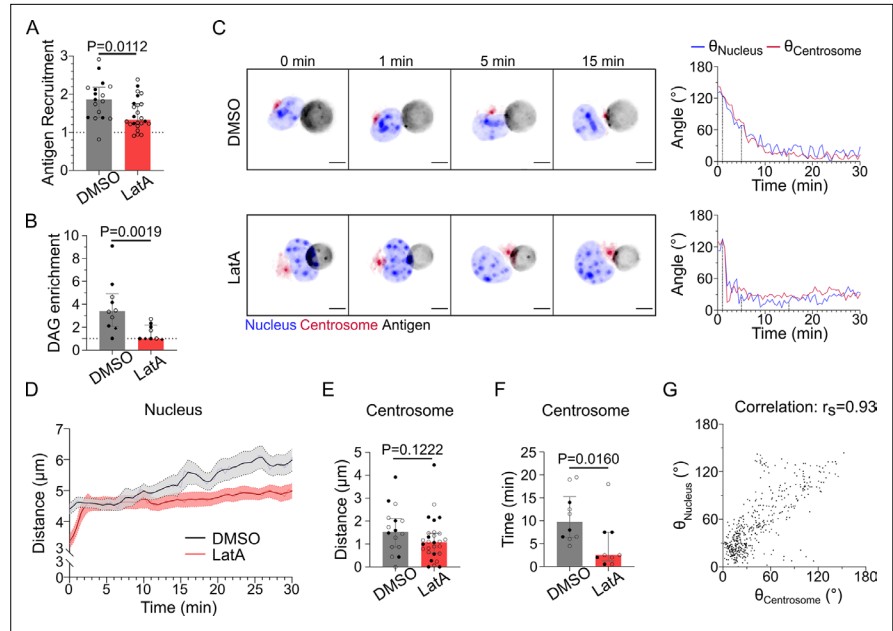

**Figure 4.** F-actin is essential for antigen recruitment and signaling amplification, but not for the establishment of the polarity axis. Experiments for this figure were performed using IIA1.6 cells, stained with SiRTubulin and Hoechst to visualize the centrosome and the nucleus, in contact with a F(ab')₂αIgG-coated droplet, imaged with SDCM 3D and quantified in 3D. Cells were pretreated for 1 hr either with DMSO or with latrunculin A 2 μM, kept in solution during the experiment. (**A**) Plateau of antigen recruitment (average values 25–30 min). Line at antigen recruitment = 1 (uniform fluorescence on the droplet). Median ± IQR, DMSO N = 7;10, LatA N = 6;18, two independent experiments, Mann–Whitney test (quantification: see *Figure 1D*). (**B**) Maximum diacylglycerol (DAG) enrichment (in 0–10 min). Median ± IQR, DMSO N = 1;5;4, LatA N = 2;5;2, three independent experiments, Mann–Whitney test (quantification: see *Figure 2A*). (**C**) Time-lapse images of untreated (DMSO) or LatA-treated cells, centrosome in red, nucleus in blue, and antigen in gray. Scale bar 5 μm. Right: angle between the cell–droplet axis and the cell–nucleus invagination (blue) or cell–centrosome (red) axis in time (quantification: see *Figure 3A*). (**D**) Nucleus–droplet distance over time. Mean ± SEM, DMSO N = 7;10, LatA N = 15;17, two independent experiments. (**E**) Average centrosome–droplet distance (25–30 min). Median ± IQR, DMSO N = 6;10, LatA N = 11;17, two independent experiments, Mann–Whitney test. (**F**) Time of centrosome polarization (threshold distance <2 μm). Median ± IQR, DMSO N = 4;6, LatA N = 4;5, two independent experiments, Mann–Whitney test. (**G**) Nucleus orientation and centrosome orientation (quantification: see *Figure 3A*) during the first 15 min, for DMSO-treated cells. N = 6;10 cells, one image every 30 s, two independent experiments. Nonparametric Spearman correlation between nucleus–centrosome pairs of data, average correlation 0.93, confidence interval: 0.86–0.97.

The online version of this article includes the following source data and figure supplement(s) for figure 4:

**Figure supplement 1.** Myosin II merely regulates antigen recruitment and diacylglycerol (DAG) signaling.

**Source data 1.** Data tables related to graphs in *Figure 4*.

**Figure supplement 1—source data 1.** Data tables related to graphs in *Figure 4—figure supplement 1*.

---

of processes: (1) 'early processes' localized at the immune synapse, such as the strong polymerization of F-actin, antigen clustering, and signaling downstream of BCR engagement, which take place in the first 3 min; and (2) global rearrangements resulting in the reorientation of the centrosome, Golgi apparatus, and nuclear invagination to the immune synapse, the recruitment of lysosomes, and later on, the rearward transport of the nucleus. These local and global events associated to synapse formation will be referred to as early and late events from now on.

## The actin cytoskeleton is needed for early but not late events of synapse formation

Having identified the temporal sequence of trafficking events associated to immune synapse formation, we next investigated their interdependency and coordination by the actin and microtubule cytoskeletons. We found that inhibition of actin polymerization with latrunculin A drastically impaired

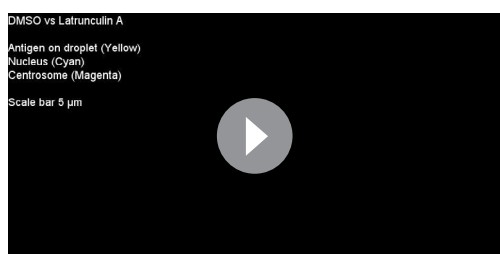

**Video 4.** Centrosome (SiRTubulin staining) and nucleus (Hoechst staining) in IIA1.6 cells treated with DMSO or latrunculin A.

https://elifesciences.org/articles/78330/figures#video4

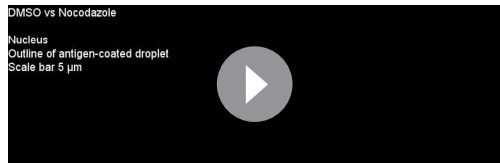

**Video 5.** Nucleus (Hoechst staining) in IIA1.6 cells treated with DMSO or nocodazole; droplet outline.

https://elifesciences.org/articles/78330/figures#video5

the clustering of antigen at the droplet surface (*Figure 4A*), as well as the production of DAG downstream of BCR signaling (*Figure 4B*). However, neither inhibition nor activation of myosin II contractility (using the inhibitor para-nitroBlebbistatin or the TRPML1 Calcium channel agonist MLSA1; *Bretou et al., 2017*; *Kumari et al., 2019*) strongly affected antigen clustering (*Figure 4—figure supplement 1A*) or DAG production (*Figure 4—figure supplement 1B and C*) at initial or late timepoint. Taken together, these results stress the importance of F-actin organization – but not actomyosin contractility – in early local events of immune synapse formation, namely, antigen clustering and BCR signaling. Interestingly, imaging centrosome and nucleus repositioning to the synapse revealed that in the absence of F-actin these global polarization processes were preserved and did even take place faster (*Figure 4C–F*, *Video 4*). This acceleration in centrosome polarization might result from loss of F-actin-dependent tethering of this organelle to the nucleus in latrunculin A-treated cells. Indeed, we previously showed that this pool of F-actin must be cleared for the centrosome to move toward the immune synapse (*Obino et al., 2016*). We observed that the centrosome faces the nuclear invagination throughout immune synapse formation, and that they reorient together to ultimately face the immune synapse independently of F-actin (*Figure 4C*). This was confirmed by the strong correlation between centrosome and nucleus orientation with respect to the cell–droplet axis (*Figure 4G*). These findings suggest that the centrosome and the nucleus reorient together, which is not affected by F-actin depolymerization. We conclude that the actin cytoskeleton is essential for the local, early events (antigen clustering and DAG production downstream of BCR signaling) of synapse formation, but not for the global, late ones (centrosome and nucleus polarization).

## The microtubule cytoskeleton controls both local and global events of synapse formation

Having established how F-actin impacts immune synapse formation, we next addressed its dependency on the microtubule cytoskeleton. For this, we treated cells with nocodazole to depolymerize microtubules. As expected, microtubule depolymerization prevented centrosome polarization (*Figure 5A*). Nucleus polarization was also impaired (*Figure 5B*). These findings are consistent with these two organelles repositioning together, as described above, and further

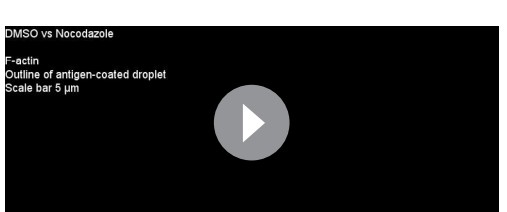

**Video 6.** F-actin in IIA1.6 cells treated with DMSO or nocodazole; droplet outline.

https://elifesciences.org/articles/78330/figures#video6

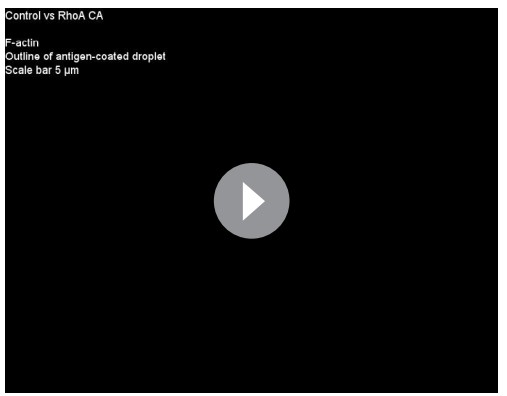

**Video 7.** F-actin in IIA1.6 cells expressing an empty vector (pRK5) or RhoA CA; droplet outline.

https://elifesciences.org/articles/78330/figures#video7

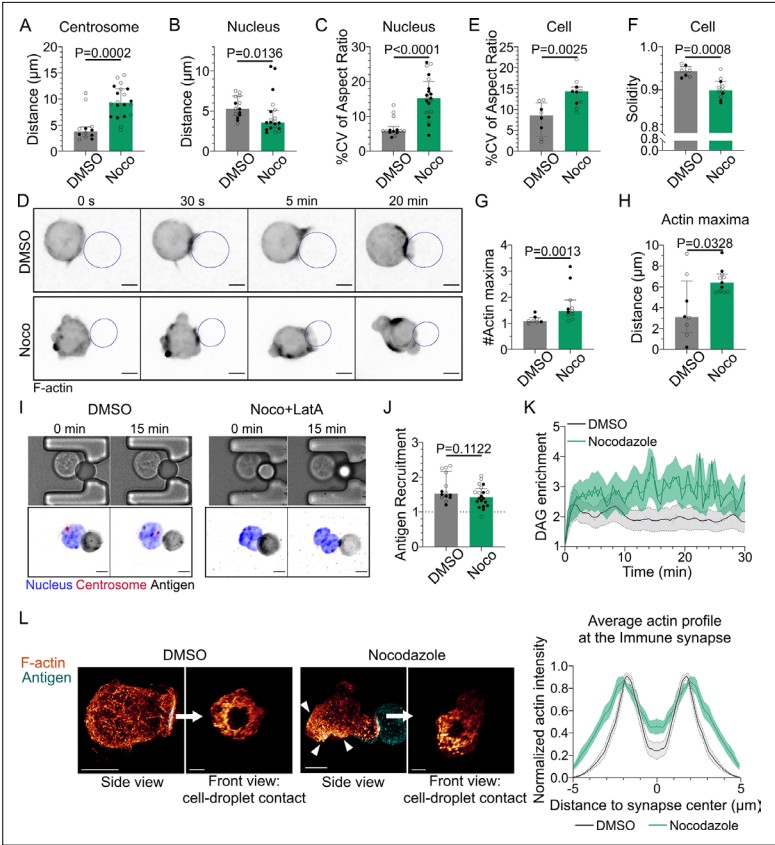

**Figure 5.** Microtubule disruption leads to intense cell and nucleus deformation, and impairs the establishment and maintenance of a polarized organization. Experiments for this figure were performed using IIA1.6 cells in contact with a F(ab')₂αIgG-coated droplet, and 3D SDCM time-lapse imaging. Cells were pretreated for 1 hr either with DMSO or with nocodazole 5 µM, kept in solution during the experiment. (**A**) Average centrosome–droplet distance (25–30 min) (median ± IQR, DMSO N = 5;7, Noco N = 9;11, two independent experiments, Mann–Whitney test). Measured in 3D using eGFP-Centrin1-expressing cells. (**B**) Average nucleus–droplet distance (25–30 min), measured in 3D, and (**C**) % coefficient of variation of 2D aspect ratio of individual nuclei over time, measured on maximum z-projections of 3D movies (median ± IQR, DMSO N = 6;8, Noco N = 12;8, two independent experiments, Mann–Whitney test). Staining: Hoechst. (**D**) Time-lapse images of F-tractin-tdTomato-expressing cells treated with DMSO or nocodazole, droplet outlined in blue. Scale bar 5 µm. (**E**) % coefficient of variation of 2D aspect ratio of individual cells over time and (**F**) median 2D solidity of individual cells (median ± IQR, DMSO N = 3;5, Noco N = 4;7, two independent experiments, Mann–Whitney test). Measured using a mask of F-tractin-tdTomato on maximum z-projections of 3D movies. (**G**) Average number of F-actin maxima detected per cell over time and (**H**) average distance of maxima to the droplet surface (median ± IQR, DMSO N = 3;5, Noco N = 4;7, two independent experiments, Mann–Whitney test). Measured on maximum z-projections of 3D movies. (**I**) Example images of untreated (DMSO) or treated (Noco 5 µM + LatA 2 µM) cells, bright-field and fluorescence (eGFP-Cent1, Hoechst, antigen). Scale bar 5 µm. (**J**) Plateau of antigen recruitment on the droplet (average values 25–30 min) (median ± IQR, DMSO N = 6;8, Noco N = 12;8, two independent experiments, Mann–Whitney test) (quantification: see *Figure 1D*). (**K**) Diacylglycerol (DAG) enrichment over time (mean ± SEM, DMSO N = 6;7, Noco N = 4;6, two independent experiments). Measured using cells expressing the DAG reporter (C1δ-GFP) (quantification: see *Figure 2A*). (**L**) Left: examples of 3D SIM immunofluorescence imaging of F-actin (phalloidin staining) and antigen on the droplet after 15–20 min of immune synapse formation. White arrowheads: sites of actin enrichment outside of the immune synapse. Side view: scale bar 5 µm. Front view: scale bar 2 µm. MIP visualization. Right: profiles of F-actin at the immune synapse, from symmetric radial scans of the immune synapse, normalized to the maxima (mean ± SEM, one representative experiment, DMSO N = 12, Noco N = 8).

The online version of this article includes the following source data for figure 5:

**Source data 1.** Data tables related to graphs in *Figure 5*.

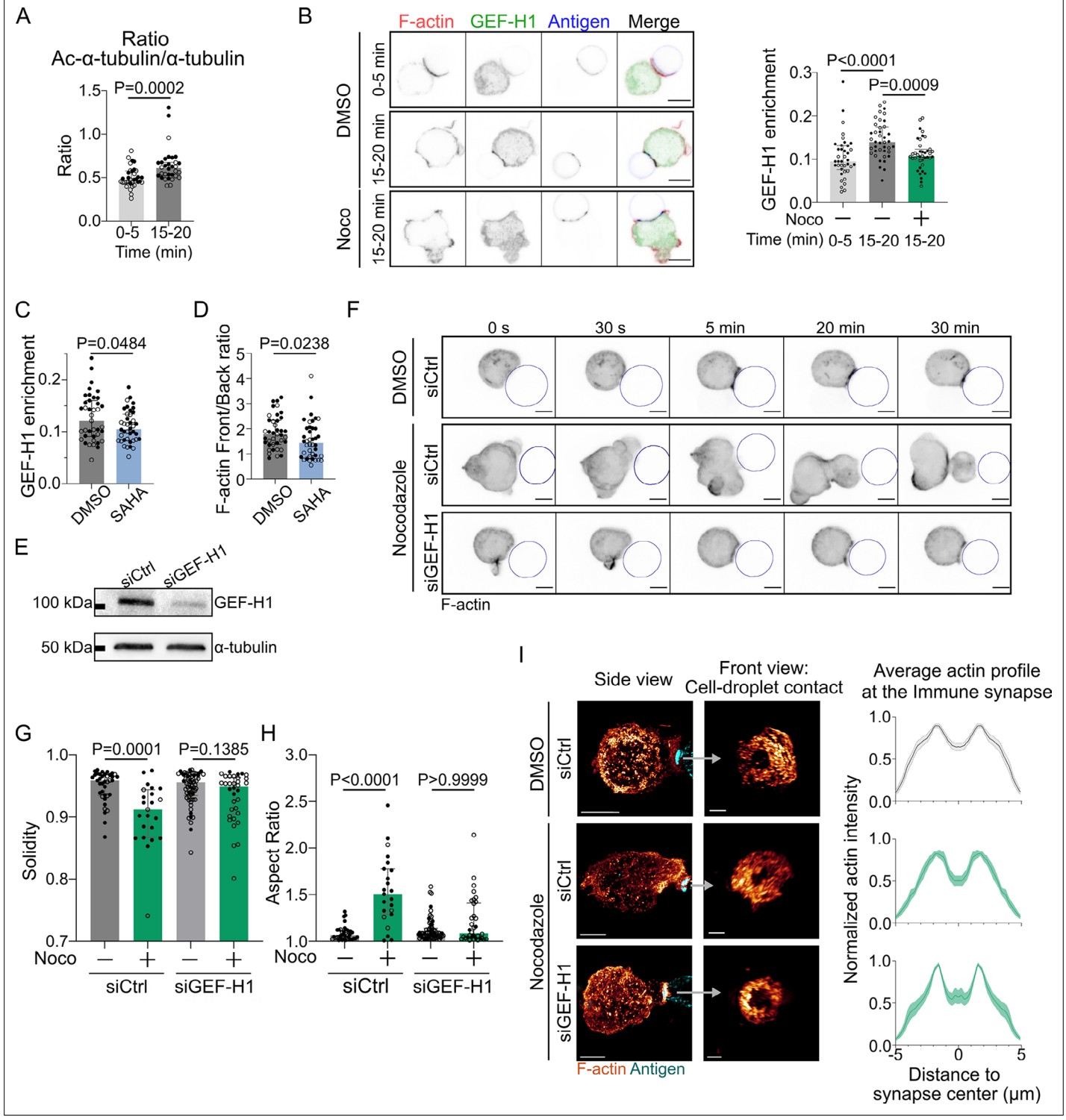

**Figure 6.** GEF-H1 is responsible for cell shape and actin patterning defects upon microtubule depletion. Experiments for this figure were performed using IIA1.6 cells transfected either with siCtrl or siGEF-H1 siRNAs 60 hr before experiment, with F-tractin-tdTomato the day before experiment, then put in contact with a F(ab')₂αIgG-coated droplet. Cells were pretreated for 1 hr with DMSO, suberoylanilide hydroxamic acid (SAHA) 10 µM or with nocodazole 5 µM, kept in solution during the experiment. (**A**) Quantification of the ratio of acetylated α-tubulin/α-tubulin in the whole cell, for IIA1.6 cells in contact with a droplet for different times, by immunofluorescence. Imaging by confocal microscopy (median ± IQR, 0–5 min N = 14;20, 15–20 min N = 18;14, two independent experiments, Mann–Whitney test). (**B**) Left: immunofluorescence images of IIA1.6 cells treated with DMSO or nocodazole, and in contact with a droplet for 0–5 min or 15–20 min. F-actin stained with phalloidin (red), GEF-H1 (green), and antigen on droplet (blue). Scale

*Figure 6 continued on next page*

*Figure 6 continued*

bar 6 µm. Right: quantification of the enrichment in GEF-H1 within 1 µm of the droplet divided by the total intensity in the cell in one plane, imaged by laser scanning confocal microscopy (LSCM), for IIA1.6 cells in contact with a droplet for different times, by immunofluorescence (median ± IQR, DMSO 0–5 min N = 20;18, DMSO 15–20 min N = 20;20, Noco 15–20 min N = 19;20, two independent experiments, Kruskal–Wallis test with multiple comparisons, Dunn's post test). (**C**) From immunofluorescence imaged with LSCM, quantification of the enrichment in GEF-H1 within 1 µm of the droplet divided by the total intensity in the cell, in one plane, and (**D**) quantification of F-actin (stained with phalloidin) on six planes (δz = 0.34 µm) around the immune synapse, ratio of intensity in the half of the cell near the synapse (front), and the half away from the synapse (back), for IIA1.6 cells treated with DMSO or SAHA in contact with a droplet for 15–20 min (median ± IQR, DMSO N = 23;16, SAHA N = 21;19, two independent experiments, Mann–Whitney test). (**E**) Western blot of GEF-H1 to evaluate the efficiency of GEF-H1 silencing. α-tubulin was used as a loading control. The blot presented is representative of two independent experiments. (**F**) Time-lapse images of F-actin in cells transfected with siCtrl or siGEF-H1 and treated with DMSO (control) or nocodazole, using SDCM 3D time-lapse imaging. Scale bar 5 µm. (**G**) Solidity in 2D and (**H**) aspect ratio of cells after 40 min of immune synapse formation (siCtrl DMSO N = 30;8, siCtrl Noco N = 19;4, siGEF-H1 DMSO N = 19;46, siGEF-H1 Noco N = 7;27, two independent experiments, Kruskal–Wallis test with multiple comparisons between DMSO and Noco, Dunn's post test), analyzed on maximum z-projections of SDCM 3D images. (**I**) Left: examples of 3D SIM immunofluorescence imaging of F-actin and antigen on the droplet after 15–20 min of immune synapse formation. Side view: scale bar 5 µm. Front view: scale bar 2 µm. MIP visualization. Right: profiles of F-actin at the immune synapse, from symmetric radial scans of the immune synapse, normalized to the maxima (mean ± SEM, pooled from two experiments, siCtrl DMSO N = 11;7, siCtrl Noco N = 5;6, siGEF-H1 Noco N = 2;7).

The online version of this article includes the following source data and figure supplement(s) for figure 6:

**Source data 1.** Raw file of the full unedited Western blot images of *Figure 6E*, and a figure with annotated images of the full Western blot.

**Source data 2.** Data tables related to graphs in *Figure 6*.

**Figure supplement 1.** Microtubules control cell shape and F-actin polarized polymerization via the GEF-H1/RhoA pathway.

**Figure supplement 1—source data 1.** Data tables related to graphs in *Figure 6—figure supplement 1*.

**Figure supplement 2.** Additional examples of 3D SIM immunofluorescence imaging of F-actin (phalloidin staining) and antigen on the droplet after 15–20 min of immune synapse formation.

suggest that their movement is driven by microtubules. Remarkably, we observed that microtubule depolymerization induced major events of nucleus and cell deformation (*Figure 5C–E*, *Videos 5 and 6*) as well as blebbing (*Figure 5F*). These deformation events were associated to aberrant F-actin distribution: multiple F-actin polymerization spots were visible all around the cell, even far from the immune synapse (*Figure 5D, G and H*). Accordingly, depolymerizing F-actin in nocodazole-treated cells with latrunculin A restored their round shape (*Figure 5I*). Microtubule depolymerization had a mild impact on antigen clustering and DAG signaling (clustering was slightly reduced while DAG was slightly more sustained) (*Figure 5J and K*). In addition, morphological analysis of the synapse showed that the stereotypical concentric actin patterning at the immune synapse was preserved (*Figure 5L*). Altogether, these results show that microtubules are instrumental for the global late events of synapse formation (centrosome and nucleus repositioning), but also suggest that microtubules maintain the polarization axis of the cell by limiting the polymerization of the actin cytoskeleton to the immune synapse, consistent with a role for these filaments in synapse maintenance.

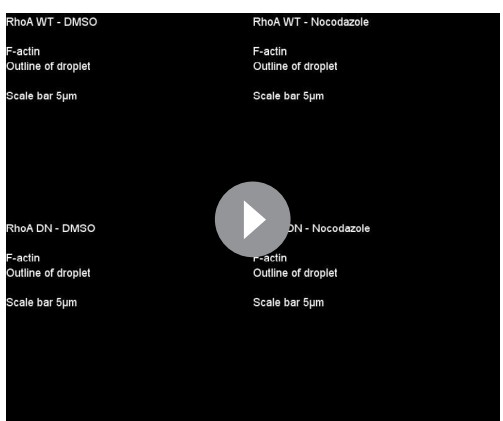

**Video 8.** F-actin in IIA1.6 cells expressing RhoA WT or RhoA DN, treated with DMSO or nocodazole; droplet outline.

https://elifesciences.org/articles/78330/figures#video8

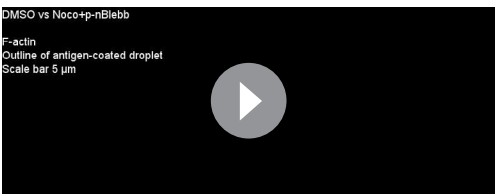

**Video 9.** F-actin in IIA1.6 cells treated with DMSO or nocodazole + para-nitroBlebbistatin; droplet outline.

https://elifesciences.org/articles/78330/figures#video9

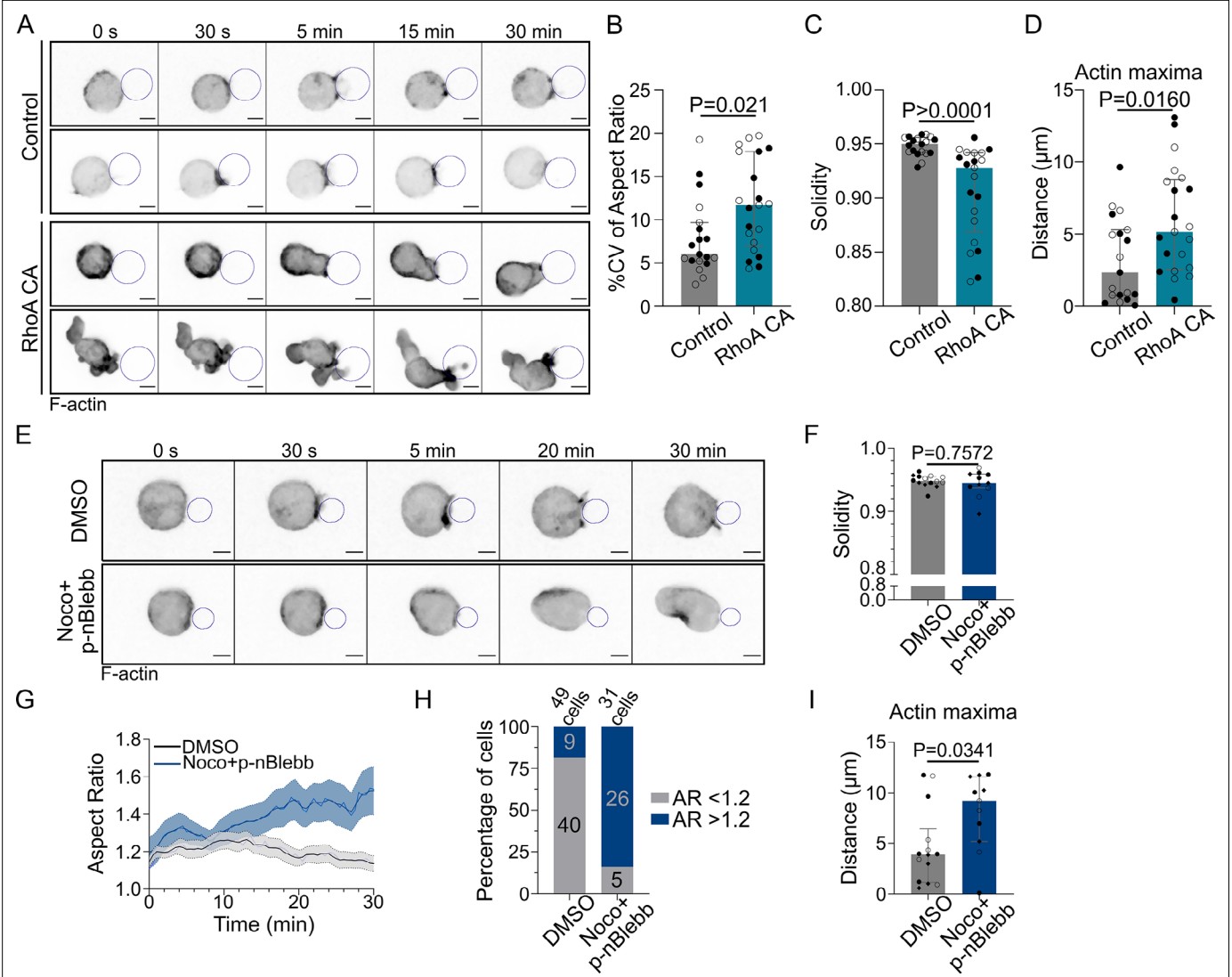

**Figure 7.** Microtubules control actin polarized polymerization via RhoA in a myosin II-independent manner. Experiments for this figure were performed using F-tractin-tdTomato-expressing IIA1.6 cells in contact with a F(ab')$_2$αIgG-coated droplet and SDCM 3D time-lapse imaging. Cells were pretreated for 1 hr either with DMSO or with nocodazole 5 µM + para-nitroBlebbistatin 20 µM, kept in solution during the experiment. (**A**) Time-lapse images of F-actin-tdTomato-expressing cells, co-transfected with either a control empty vector (pRK5) or expressing RhoA CA (constitutively active). Scale bar 5 µm. (**B**) % coefficient of variation of 2D aspect ratio of individual cells over time, (**C**) Median 2D solidity of individual cells and (**D**) average distance of actin maxima to the droplet surface (median ± IQR, Control N = 10;9, RhoA CA N = 9;12, two independent experiments, Mann–Whitney test), analyzed on maximum z-projections. (**E**) Time-lapse images of F-tractin-tdTomato-expressing cells treated with DMSO or nocodazole + p-nBlebb, droplet outlined in blue. Scale bar 5 µm. (**F**) Median 2D solidity of maximum z-projections of individual cells over time (median ± IQR, DMSO N = 5;5;4, Noco + p-nBlebb N = 4;3;4, three independent experiments, Mann–Whitney test). (**G**) Aspect ratio of z-projections of cells in time (mean ± SEM, DMSO N = 5;5;4, Noco + p-nBlebb N = 4;3;4, three independent experiments). (**H**) Percentage of cells with aspect ratio >1.2 or <1.2 after 40 min of synapse formation. (**I**) Average distance of F-actin maxima to the droplet over 30 min of synapse formation (median ± IQR, DMSO N = 5;5;4, Noco + p-nBlebb N = 4;3;4, three independent experiments, Mann–Whitney test) (quantification: as in *Figure 5H*).

The online version of this article includes the following source data and figure supplement(s) for figure 7:

**Source data 1.** Data tables related to graphs in *Figure 7*.

**Figure supplement 1.** Cell deformation upon microtubule depletion is RhoA-dependent.

**Figure supplement 1—source data 1.** Data tables related to graphs in *Figure 7—figure supplement 1*.

# Microtubules restrict actin polymerization to the immune synapse via GEF-H1 and RhoA

How do microtubules restrict actin polymerization to allow its accumulation at the immune synapse and prevent aberrant non-polarized actin distribution? A good candidate to be involved in this process is the guanine exchange factor H1 (GEF-H1), an activator of the RhoA small GTPase that is released from microtubules upon depolymerization (*Chang et al., 2008*). GEF-H1 was recently shown to be also released upon microtubule acetylation, allowing its recruitment to the B cell immune synapse (*Sáez et al., 2019*; *Seetharaman et al., 2021*). We tested that microtubules are acetylated upon BCR activation (*Figure 6A*). Accordingly, we observed that GEF-H1 accumulated at the immune synapse upon BCR engagement (*Figure 6B*). Noticeably, treatment of B cells with nocodazole or with the histone deacetylase inhibitor suberoylanilide hydroxamic acid (SAHA) (*Zhang et al., 2003*) led to a marked decrease in the synaptic fraction of GEF-H1 (*Figure 6B and C*). Actin was also found to be less polarized in SAHA-treated cells (see back/front ratio, *Figure 6D*). These results suggest that by globally enhancing GEF-H1 release both microtubule depolymerization and acetylation lead to a decrease in the relative enrichment – or polarization – of this protein at the synapse. As a consequence of this, actin polymerization now takes place all around the cell cortex, consistent with a need for microtubules to restrict the activity of GEF-H1 to the B cell immune synapse. To test this hypothesis, we silenced GEF-H1 expression using siRNA (*Figure 6E*). We found that GEF-H1 silencing normalizes most of the effects of microtubules depletion: it reduced cell deformation and blebbing (*Figure 6F–H*). Rescue experiments confirmed that the silencing was specific of this GEF (*Figure 6—figure supplement 1A and B*). Silencing GEF-H1 also slightly altered antigen recruitment, but this effect was compensated by microtubules disruption (*Figure 6—figure supplement 1C*). In microtubules-depleted cells, actin polarity was strongly perturbed while synaptic actin patterns were mildly altered. GEF-H1 silencing in nocodazole-treated cells restored both polarization (see illustrations in *Figure 6—figure supplement 1D* and axial profiles in *Figure 6—figure supplement 1E*) and synaptic actin patterns (*Figure 6I*, *Figure 6—figure supplement 2*) as observed in untreated cells. These results indicate that the aberrant non-polarized actin polymerization observed upon treatment of B lymphocytes with nocodazole most likely results from GEF-H1 release from microtubules. To further probe the role of GEF-H1, we perturbed its downstream Rho GTPase, RhoA. We found that B cells expressing a constitutively active form of RhoA (RhoA L63, referred to as RhoA CA) displayed a phenotype similar to the one of nocodazole-treated cells: aberrant non-polarized actin polymerization, dynamic cell deformation, and blebbing (*Figure 7A–D*, *Video 7*). Conversely, overexpression of a dominant negative form of RhoA (RhoA DN) prevented cell deformation and blebbing upon nocodazole treatment, similar to the effect of GEF-H1 silencing (*Figure 7—figure supplement 1A–C*, *Video 8*). These data are consistent with GEF-H1 restricting RhoA activity and actin nucleation at the B cell immune synapse.

The activation of RhoA by GEF-H1 leads to both nucleation of linear actin filaments by diaphanous formins (mDia) and activation of myosin II by the ROCK kinase for contraction of these filaments (*Watanabe et al., 1997*; *Amano et al., 1997*). We, therefore, asked whether modulation of actin nucleation or myosin II activity had any impact on the phenotype of nocodazole-treated cells. Noticeably, we found that while myosin II inhibition (using para-nitroBlebbistatin) prevented cell blebbing upon microtubule depolymerization (*Figure 7E and F*), it did not restore cell shape, with cells elongating over time (*Figure 7E, G and H*), nor polarized actin polymerization (*Figure 7E and I*, *Video 9*).

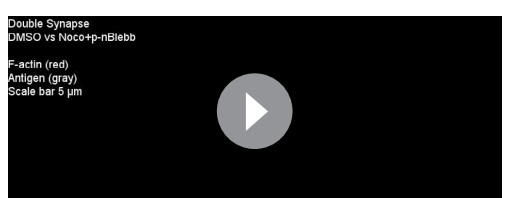

**Video 10.** F-actin in IIA1.6 cells treated with DMSO or nocodazole + para-nitroBlebbistatin, contacting two droplets. Example of a cell bringing droplets together (DMSO) and taking droplets apart (nocodazole + para-nitroBlebbistatin).
https://elifesciences.org/articles/78330/figures#video10

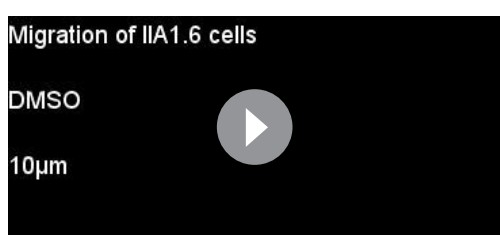

**Video 11.** Bright-field movies of migrating IIA1.6 cells treated with DMSO (Control) or nocodazole, on a BSA-coated dish, in contact with an antigen-coated droplet. Scale bar 10 µm.
https://elifesciences.org/articles/78330/figures#video11

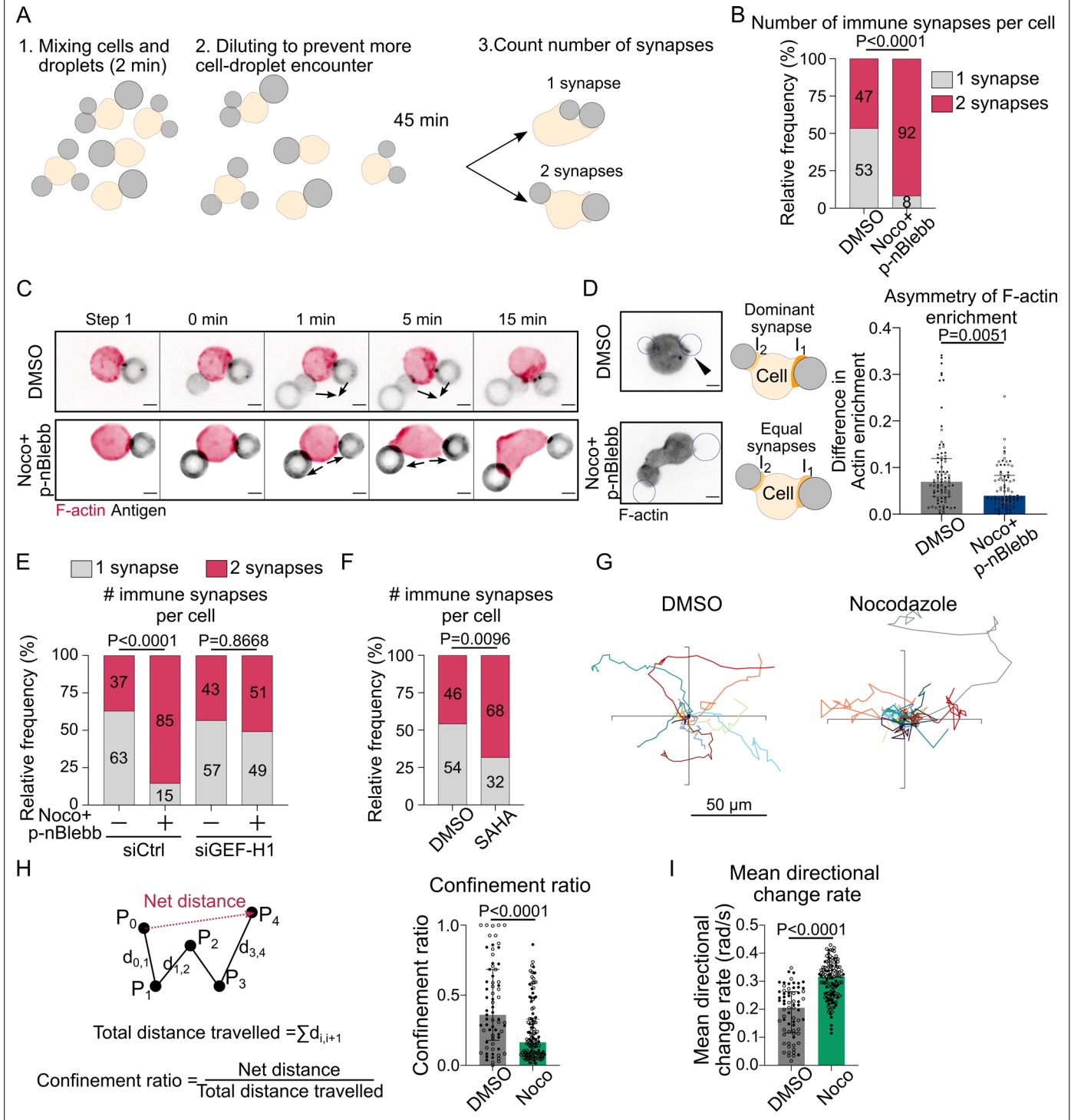

**Figure 8.** Microtubule depletion favors the formation of multiple polarity axis. Experiments for this figure were performed using F-tractin-tdTomato-expressing-IIA1.6 cells in contact with a F(ab')₂αIgG-coated droplet and SDCM 3D time-lapse imaging. Cells were pretreated for 1 hr with DMSO, with suberoylanilide hydroxamic acid (SAHA) 10 μM or with nocodazole 5 μM + para-nitroBlebbistatin 20 μM, which was kept in the media throughout experiments. (**A**) Schematic of the concept of the multiple synapse experiment. Considering only cells in contact with exactly two droplets, counting number of contact areas (number of synapses) after 45 min. (**B**) Number of immune synapses per cell treated with DMSO or Noco + p-nBlebb (DMSO N = 74;70, Noco + p-nBlebb N = 54;67, two independent experiments, Mann–Whitney test p=0.0038), from SDCM 3D imaging of cells and droplets. (**C**) Examples of time-lapse images of F-actin and antigen on the droplet. Situation of a cell (untreated) bringing droplets closer into one immune synapse, and of a cell (treated with nocodazole and para-nitroBlebbistatin) taking droplets apart. Scale bar 5 μm. 3D time-lapse SDCM imaging in the

*Figure 8 continued on next page*

*Figure 8 continued*

microfluidic chip. (**D**) Left: examples of images (from SDCM 3D time-lapse) of F-actin and antigen on the droplet. Situation of a cell (untreated) with one synapse more enriched in F-actin and of a cell (treated with nocodazole + para-nitroBlebbistatin) with equivalent synapses. Scale bar 5 μm. Right: To assess the asymmetry in F-actin enrichment between multiple synapses and the presence of a dominant, more enriched synapse, we compute here the difference of enrichment in F-actin between immune synapses, per cell (DMSO N = 44;42, Noco + p-nBlebb N = 26;50, two independent experiments, Mann–Whitney test) (quantification of F-actin enrichment: see *Figure 2B*). Quantification from SDCM 3D images in the microfluidic chip. (**E**) Number of immune synapses per cell transfected 60 hr before with siCtrl or siGEF-H1, and treated with DMSO or Noco + p-nBlebb (siCtrl DMSO N = 25;29, siCtrl Noco + p-nBlebb N = 28;34, siGEFH1 DMSO N = 24;29 siGEF-H1 Noco + p-nBlebb N = 29;26, two independent experiments, Kruskal–Wallis test with Dunn's post test for multiple comparisons), from SDCM 3D imaging of cells and droplets. (**F**) Number of immune synapses per cell treated with DMSO or suberoylanilide hydroxamic acid (SAHA) 10 μM (DMSO N = 32;27, SAHA N = 57;28, two independent experiments, Mann–Whitney test), from SDCM 3D imaging of cells and droplets. (**G**) Example trajectories of migrating IIA1.6 B lymphocytes in contact with an antigen-coated droplet, representative of two experiments, 14 trajectories per condition, 7 trajectories per experiment. Plot over 2 hr. Analysis of migration from videomicroscope bright-field time-lapse imaging. (**H**) Confinement ratio and (**I**) directional change rate of trajectories (2 hr, image every 4 min) of migrating IIA1.6 cells in contact with a droplet (DMSO N = 33;36, Noco N = 48;73, two independent experiments, Mann–Whitney test).

The online version of this article includes the following source data for figure 8:

**Source data 1.** Data tables related to graphs in *Figure 8*.

These results suggest that actin nucleation, rather than myosin II activation, downstream of GEF-H1 and RhoA activation is responsible for the non-polarized polymerization of actin upon microtubule depolymerization. Accordingly, simultaneous depolymerization of actin and microtubules prevented cell deformation, restoring both cell and nucleus shape (*Figure 5I*).

## Restriction of actin nucleation by microtubules promotes the formation of a unique immune synapse

Our results suggest that by titrating GEF-H1 microtubules tune the level of RhoA activation to restrict actin polymerization to the immune synapse, thus stabilizing a single actin polarity axis. We hypothesized that such regulatory mechanism might help B cells maintaining a unique immune synapse, rather than forming multiple synapses all over their cell body. To test this hypothesis, we put cells in contact with several droplets within a few minutes and observed how they would interact (*Figure 8A*). For this experiment, we chose to use cells treated with both nocodazole and para-nitroBlebbistatin to prevent excessive blebbing and facilitate the analysis. We observed two types of cell behaviors: they either brought the droplets together into a single immune synapse or formed multiple, separated immune synapses (*Figure 8A*). Noticeably, microtubule-depleted cells formed more multiple separated synapses than control cells (*Figure 8B*). Accordingly, while control cells were able to merge contacted droplets into a unique immune synapse, this was not observed in cells whose microtubule were depolymerized (*Figure 8C*, *Video 10*). These results are consistent with microtubule being required for the formation and maintenance of a unique immune synapse, wherein F-actin polymerization concentrates, rather than multiple dispersed ones.

To test this hypothesis, we computed the difference between the synapses in terms of F-actin enrichment on the subpopulation of cells that formed two spatially separated immune synapses with two droplets (to facilitate imaging and quantification, this experiment was performed in the microfluidic chip). We found that, while control cells tend to have a stronger F-actin enrichment at one synapse, indicating that they are able to establish and maintain a dominant polarity axis, this was less often observed in nocodazole-treated cells (*Figure 8D*). Remarkably, GEF-H1 silencing in nocodazole-treated cells led to the re-establishment of a single polarity axis (*Figure 8E*). The role of GEF-H1 in controlling the uniqueness of the polarity axis was further reinforced by the observation of multiple synapses in SAHA-treated cells (*Figure 8F*), in which GEF-H1 polarized accumulation was compromised. The capacity of establishing and maintaining a single polarity axis is essential for cells to migrate in a directional manner (*Maiuri et al., 2015*), which might be required for activated B lymphocytes to reach the border of the T cell zone in lymph nodes for T-B cooperation. We therefore hypothesized that by compromising the polarity axis of B cells microtubule depletion might also impair their migratory capacity. To test this hypothesis, we plated B lymphocytes on BSA-coated surfaces after incubation with antigen-coated droplets. We found that B cells whose microtubules had been depolymerized with nocodazole exhibited more confined trajectories as compared to untreated cells (*Figure 8G and H*, *Video 11*). Consistently, nocodazole-treated cells exhibit larger orientation

change at each step (mean directional change rate or angular velocity) (*Figure 8I*), indicating that their migration is less directional than the one of control B lymphocytes. Altogether, our results strongly suggest that, by restricting RhoA-dependent actin polymerization via GEF-H1, microtubules allow the maintenance of a single polarity axis and stabilize in space and time a unique immune synapse in B lymphocytes. We propose that this process helps B cells properly extracting, processing, and presenting antigens to T lymphocytes.

## Discussion

In this work, we used a custom microfluidic system to study the coordination by actin and microtubule cytoskeletons of the various events associated to immune synapse formation in B lymphocytes. We observed that this process is characterized by two classes of events: a first phase (in the first 3.5 min), where F-actin is strongly polymerized at the site of contact, leading to antigen accumulation and production of DAG as a result of BCR signaling, and a second phase during which the centrosome is reoriented toward the immune synapse together with the Golgi apparatus and lysosomes while the nucleus undergoes a rotation followed by backward transport. The timescales we found for late polarization events are shorter than the ones measured for B cells in other systems (e.g., centrosome polarized in 30 min [*Yuseff et al., 2011*], nucleus fully polarized in 30 min [*Ulloa et al., 2022*], lysosomes maximally clustered in 40 min [*Spillane and Tolar, 2018*]) and much closer to results found in T cells (*Gawden-Bone et al., 2018*; *Yi et al., 2013*; *Hooikaas et al., 2020*), possibly due to the properties of the substrate that we used for antigen presentation. We found that F-actin polymerization is only needed for the first phase, in contrast to microtubules that not only control centrosome and organelle repositioning, but further maintain a unique polarity axis by restricting actin nucleation to the immune synapse. We propose that this mechanism reinforces a single synapse and guarantees B cell persistent migration to the T cell zone for cooperation with T lymphocytes.

How do microtubules restrict F-actin polymerization to the immune synapse? We identified GEF-H1 as a key player in this process, which limits RhoA activity and downstream actin nucleation to the synapse. Indeed, we observed that global activation of the GEF-H1-RhoA axis induced actin polymerization outside of the synapse, independently of myosin II activity. Interestingly, it was recently shown that microtubules were acetylated in the vicinity of the centrosome upon immune synapse formation, resulting in the local release and activation of GEF-H1 (*Sáez et al., 2019*; *Seetharaman et al., 2021*). Our results suggest that GEF-H1 might activate RhoA to trigger downstream formin-dependent actin nucleation at the immune synapse exclusively. In this model, RhoA would remain inactive in the rest of the cell, most likely due to GEF-H1 trapping on microtubules deacetylated by HDAC6 (*Hubbert et al., 2002*; *Seetharaman et al., 2021*). Indeed, inhibition of microtubule deacetylation decreases the polarization of GEF-H1 to the synapse, leading to uncontrolled actin polymerization all over the cell cortex. We suggest that this 'local activation' of GEF-H1 and 'global inhibition' by trapping on deacetylated microtubules is reminiscent of the Local Excitation Global Inhibition model described in amoebas, where symmetry breaking arises from and is stabilized by a local positive feedback (PIP3 that promotes F-actin polymerization) combined to a globally active diffusible inhibitory signal (PTEN, a PIP3 phosphatase) (*Parent and Devreotes, 1999*; *Janetopoulos et al., 2004*; *Devreotes and Janetopoulos, 2003*). Of note, this model suggests that histone deacetylase inhibitors (some of them already used as drugs against autoinflammatory diseases; *Licciardi and Karagiannis, 2012*; *Bodas et al., 2018*; *Nijhuis et al., 2019*) could, by impairing polarization of B cells toward the synapse, prevent hyper activity of immune cells in pathological conditions, such as lymphoma or autoimmune diseases.

The Local Excitation Global Inhibition model predicts the establishment of a single stable polarity axis. Accordingly, our experiments show that unperturbed B lymphocytes favor the formation of a unique synapse over multiple ones, even when particulate antigens are presented from several locations. We propose that this mechanism, at least in enzymatic extraction, could help improving antigen extraction. Indeed, GEF-H1 has been shown to be necessary for the assembly of the exocyst complex at the immune synapse, and therefore for protease secretion (*Sáez et al., 2019*). In this context, the localized release and activation of GEF-H1 by microtubules at the immune synapse might allow for the concentration of resources, promoting F-actin polymerization and optimizing proteolytic extraction at one unique site. Polarization of the centrosome and reorientation of the microtubule network would thus reduce the dispersion of resources in secondary synapses. Indeed, the release of proteases in

several locations or in an open environment (as opposed to the tight synaptic cleft) could result in a lower local concentration of proteases, and therefore lower the efficiency of antigen uptake. A unique polarity axis could also help T/B cooperation as antigen-loaded B cells must migrate to the T cell zone for antigen presentation to T lymphocytes, and, as here shown, their capacity to migrate directionally depends on the robustness of cell polarity (see also *Carrasco and Batista, 2007*; *Maiuri et al., 2015*). In addition, it has been shown that B cells can undergo asymmetric cell division upon synapse formation and antigen extraction, which prevents antigen dilution upon cell division, an event that also requires a stable polarity axis (*Thaunat et al., 2012*; *Sawa, 2012*). Future experiments aimed at studying how these downstream events of synapse formation are regulated when B cells nucleate actin all over their cell cortex and form multiple contacts should help address these questions.

In conclusion, we showed that microtubules can act as a master regulator of actin polymerization, maintaining the formation of a single immune synapse in B lymphocytes. This control relies on the GEF-H1-RhoA axis, which may be at the core of a 'Local Excitation Global Inhibition' model. Our work points at the interaction between actin and microtubules as a way to control the axis of cell polarity that might be common to a larger class of cells.

# Materials and methods

## Key resources table

| Reagent type (species) or resource | Designation | Source or reference | Identifiers | Additional information |
|---|---|---|---|---|
| Cell line (*Mus musculus*) | IIA1.6 | *Yuseff et al., 2011* | Cellosaurus A20.IIA (CVCL_0J27) | IgG$^+$ B lymphoma cell line |
| Genetic reagent (*M. musculus*) | LifeAct-GFP mice/C57BL/B6 | *Riedl et al., 2008* | MGI:4831036 | |
| Software, algorithm | Fiji | *Schindelin et al., 2012* | | https://imagej.net/Fiji |
| Software, algorithm | Icy bioimage analysis | *de Chaumont et al., 2012* | | https://icy.bioimageanalysis.org/ |
| Software, algorithm | MATLAB | MathWorks | | |
| Software, algorithm | GraphPad PRISM | GraphPad Software | Version 9.2.0 | |
| Software, algorithm | RStudio | RStudio | | |
| Software, algorithm | Metamorph | Molecular Devices | | |
| Software, algorithm | SoftWoRx | Image Precision | | |
| software, algorithm | Imaris Viewer | Imaris | | |
| Sequence-based reagent | ON-TARGETplus Control n=Non-Targeting Pool | Dharmacon | D-001810-10-05 | |
| Sequence-based reagent | SMARTPool ON-TARGETplus Mouse Arhgef2 siRNA | Dharmacon | L-040120-00-0005 | |
| Commercial assay or kit | B cell isolation kit | Miltenyi | 130-090-862 | |
| Commercial assay or kit | LS columns | Miltenyi | 130-042-401 | |
| Commercial assay or kit | 10µL Neon Transfection system | Thermo Fisher | MPK1096 | 1300 V, 20 ms, two pulses |
| Commercial assay or kit | Amaxa Nucleofector kit R | Lonza | VCA-1001 | T-016 program |
| Chemical compound, drug | DSPE-PEG(2000) | Avanti Lipids, Coger | 880129-10mg | Resuspended in chloroform |
| Chemical compound, drug | Soybean oil | Sigma-Aldrich | CAS# 8001-22-7 | |

*Continued on next page*

*Continued*

| Reagent type (species) or resource | Designation | Source or reference | Identifiers | Additional information |
|---|---|---|---|---|
| Chemical compound, drug | Pluronic F68 | Sigma-Aldrich | CAS# 9003-11-6 | |
| Chemical compound, drug | Sodium alginate | Sigma-Aldrich | CAS# 9005-38-3 | |
| Chemical compound, drug | Tween 20 | Sigma-Aldrich | CAS# 9005-64-5 | |
| Chemical compound, drug | $Na_2HPO_4 \cdot 7H_2O$ | Merck | CAS# 7782-85-6 | |
| Chemical compound, drug | $NaH_2PO_4 \cdot H_2O$ | Carlo Erba | CAS# 10049-21-5 | |
| Chemical compound, drug | Streptavidin Alexa Fluor 405 | Thermo Fisher | S32351 | |
| Chemical compound, drug | Streptavidin Alexa Fluor 488 | Thermo Fisher | S11223 | |
| Chemical compound, drug | Streptavidin Alexa Fluor 546 | Thermo Fisher | S11225 | |
| Chemical compound, drug | Streptavidin Alexa Fluor 647 | Thermo Fisher | S32357 | |
| Chemical compound, drug | Biotin labeled bovine albumin | Sigma-Aldrich | A8549-10MG | |
| Chemical compound, drug | PDMS-RTV 615 | Neyco | RTV615 | 1:9 ratio |
| Chemical compound, drug | PVP K90 | Sigma-Aldrich | 81440 | 0.2 % $\frac{w}{v}$ in MilliQ water |
| Chemical compound, drug | Latrunculin A | Abcam | ab144290 | 2 µM, 1 hr |
| Chemical compound, drug | para-nitroBlebbistatin | Optopharma | 1621326-32-6 | 20 µM, 1 hr |
| Chemical compound, drug | Nocodazole | Sigma | M1404 | 5 µM, 1 hr |
| Chemical compound, drug | MLSA1 | Tocris | 4746 | 1 µM, 1 hr |
| Chemical compound, drug | SAHA | Tocris | 4652 | 10 µM, 1 hr |
| Chemical compound, drug | Hoechst 33342 | Thermo Fisher | R37605 | |
| Chemical compound, drug | LysoTracker Deep Red | Thermo Fisher | L12492 | Cell labeling 50 nM, 45 min |
| Chemical compound, drug | SiRTubulin kit | Spirochrome AG, Tebu-bio | SC002 | 100 nM SiRTubulin + 10 µM verapamil |
| Other | Tygon Medical Tubing | Saint-Gobain (VWR) | ND 100-80 | Tubing for injection in microfluidic chips (see 'Live imaging of IIA1.6 cell polarization in microfluidic chips') |
| Other | Stainless steel dispensing needles 23GA | Kahnetics | KDS2312P | Needle for injection in microfluidic chips (see 'Live imaging of IIA1.6 cell polarization in microfluidic chips') |
| Antibody | Anti-B220 AF647 (rat monoclonal) | BioLegend | 103229 | On live cells (1:100), incubation 15 min at 4°C |

*Continued on next page*

*Continued*

| Reagent type (species) or resource | Designation | Source or reference | Identifiers | Additional information |
|---|---|---|---|---|
| Antibody | Biotin-SP-conjugated F(ab′)₂ goat polyclonal anti-mouse IgG | Jackson ImmunoResearch | 115-066-072 | Droplet functionalization (5.7 µL) |
| Antibody | Biotin-SP-conjugated F(ab′)₂ goat polyclonal anti-mouse IgM | Jackson ImmunoResearch | 115-066-020 | Droplet functionalization (5.7 µL) |
| Antibody | Anti-EXOC7 (rabbit polyclonal) | abcam | ab95981 | IF (1:200) |
| Antibody | Anti-GEF-H1 (rabbit polyclonal) | abcam | ab155785 | WB (1:1000), IF (1:100) |
| Antibody | Anti-α-tubulin (rat monoclonal) | Bio-Rad | MCA77G | WB (1:1000), IF (1:1000) |
| Antibody | Anti-acetyl-α-tubulin (Lys40) (rabbit monoclonal) | Cell Signaling | 5335 | IF (1:250) |
| Recombinant DNA reagent | eGFP-Centrin1 | *Obino et al., 2016* | | |
| Recombinant DNA reagent | C1δ-GFP | *Botelho et al., 2000* | | |
| Recombinant DNA reagent | GEF-H1 | Origene | RG204546 | |
| Recombinant DNA reagent | pRK5myc RhoA L63 | Addgene, *Nobes and Hall, 1999* | 15900 | |
| Recombinant DNA reagent | RhoA WT EGFP | *Subauste et al., 2000* | | |
| Recombinant DNA reagent | RhoA T19N EGFP | *Subauste et al., 2000* | | |

## Cells and cell culture

The mouse IgG⁺ B lymphoma cell line IIA1.6 (derived from the A20 cell line [ATCC# TIB-208], listed in Cellosaurus as A20.IIA CVCL_0J27) was cultured as previously reported (*Yuseff et al., 2011*) in CLICK Medium (RPMI 1640 – GlutaMax-I + 10% fetal calf serum, 1% penicillin–streptomycin, 0.1% β-mercaptoethanol, and 2% sodium pyruvate). Fetal calf serum was decomplemented for 40 min at 56°C. All cell culture products were purchased from Gibco/Life Technologies. All experiments were conducted in CLICK + 25 mM HEPES (15630080, Gibco). The cell line was confirmed to be free of mycoplasma contamination. The transgenic LifeAct-GFP mouse line has been described elsewhere (*Riedl et al., 2008*) and was kept in the C57BL/B6 background. The experiments were performed on 8–12-week-old male or female mice. Animal care conformed strictly to European and French national regulations for the protection of vertebrate animals used for experimental and other scientific purposes (Directive 2010/63; French Decree 2013-118). Mature splenic B lymphocytes were purified using the MACS B cell isolation kit (Miltenyi, 130-090-862, with LS columns Miltenyi, 130-042-401). Primary B cells were kept in CLICK Medium + 25 mM HEPES + 1× non-essential amino acids (NEAA, Gibco, 11140050).

## Antibodies and reagents

### For droplet preparation fabrication and functionalization

DSPE-PEG(2000) biotin in chloroform (Avanti Lipids, Coger 880129C-10mg), soybean oil (Sigma-Aldrich, CAS# 8001-22-7), Pluronic F68 (Sigma-Aldrich, CAS# 9003-11-6), sodium alginate (Sigma-Aldrich, CAS# 9005-38-3), Tween 20 (Sigma-Aldrich, CAS# 9005-64-5), Na₂HPO₄ 7H₂O (sodium phosphate dibasic heptahydrate, M = 268 g/mol, Merck, CAS# 7782-85-6), NaH₂PO₄ H₂O (sodium phosphate monobasic monohydrate M = 138 g/mol, Carlo Erba, CAS# 10049-21-5), streptavidin Alexa Fluor 488 (Thermo Fisher, S11223), streptavidin Alexa Fluor 546 (Thermo Fisher S11225), streptavidin Alexa Fluor 647 (Thermo Fisher S32357), streptavidin Alexa Fluor 405 (Thermo Fisher S32351),

biotin-SP-conjugated AffiniPure F(ab')$_2$ Fragment Gt anti-Ms IgG (Jackson ImmunoResearch 115-066-072), biotin labeled bovine albumin (Sigma-Aldrich A8549-10MG), and biotin-SP-conjugated Affini-Pure F(ab')$_2$ Fragment Gt anti-Ms IgM (Jackson ImmunoResearch 115-066-020).

## For microfluidic chips

PDMS-RTV 615 (Neyco RTV6115), polyvinylpyrrolidone K90 (Sigma 81440, called PVP), Medical tubing, Tygon ND 100-80 (Saint-Gobain), stainless steel plastic hub dispensing needles 23 GA (Kahnetics KDS2312P), and FluoroDish (World Precision Instruments FD35).

## Dyes and plasmids for live-cell imaging

Hoechst 33342 (Thermo Fisher, R37605) kept in solution, LysoTracker Deep Red (Thermo Fisher, L12492) 50 nM in incubator for 45 min, then wash, SiRTubulin kit (Spirochrome AG, Tebu-bio SC002) 100 nM SiRTubulin + 10 µM verapamil >6 hr, rat anti-B220/CD45R AF 647 (BioLegend, 103229) 1:100, 15 min at +4°C, then washed and resuspended in media, eGFP-Centrin1 plasmid used in *Obino et al., 2016*, F-tractin tdTomato obtained from the team of Patricia Bassereau (Institut Curie, Paris), Rab6-mCherry plasmid obtained from Stéphanie Miserey (Institut Curie, Paris), and C1δ-GFP plasmid was obtained from Sergio Grinstein (*Botelho et al., 2000*). GEF-H1 (ARHGEF2) (NM_004723) Human Tagged ORF Clone in pCMV6-AC-GFP vector was bought from Origene (RG204546). pRK5myc RhoA L63 (RhoA CA – constitutively active) was a gift from Alan Hall (Addgene plasmid 15900; http://n2t.net/addgene:15900; RRID:Addgene_15900) (*Nobes and Hall, 1999*), and an empty pRK5myc vector was used as a negative control. RhoA WT EGFP and RhoA T19N EGFP (RhoA DN – dominant-negative) were a gift from Matthieu Coppey's lab (*Subauste et al., 2000*). Expression of Ftractin-tdTomato, Rab6-mCherry, C1δ-GFP, pRK5myc, and RhoA L63 was achieved by electroporating 1×10$^6$ B lymphoma cells with 0.25–0.5 µg of plasmid using the 10 µL Neon Transfection system (Thermo Fisher). Expression of RhoA WT and RhoA T19N was achieved by electroporating 1×10$^6$ B lymphoma cells with 3 µg of plasmid using the 10 µL Neon Transfection system (Thermo Fisher). Expression of pRK5 or GEF-H1 for experiments of rescue of silencing was achieved by electroporating 1×10$^6$ B lymphoma cells with 1.5 µg of plasmid using the 10 µL Neon Transfection system (Thermo Fisher), the night before the experiment. Expression of eGFP-Centrin1 was achieved by electroporating 4×10$^6$ B lymphoma cells with 4 µg of plasmid using the Amaxa Cell Line Nucleofector Kit R (T-016 program, Lonza). Cells were cultured in CLICK Medium for 5–16 hr before imaging.

For siRNA silencing, IIA1.6 cells were transfected 60–70 hr before live experiment with 40 pmol siRNA per 10$^6$ cells using the 10 µL Neon Transfection system (Thermo Fisher) and ON-TARGETplus Control n=Non-Targeting Pool (Dharmacon, D-001810-10-05) or SMARTPool ON-TARGETplus Mouse Arhgef2 siRNA (Dharmacon, L-040120-00-0005).

## For immunofluorescence and Western blot

Formaldehyde 16% in aqueous solution (Euromedex, 15710), BSA (Euromedex, 04-100-812-C), PBS (Gibco, 10010002), rabbit anti-EXOC7 (abcam, ab95981, 1/200 for IF), rabbit anti-GEF-H1 (abcam, ab155785, 1/1000 for WB, 1/100 for IF), rat anti-α-tubulin (Bio-Rad, MCA77G, 1/1000 for WB and IF), rabbit anti-acetyl-α-tubulin (Lys40) (D20G3) (Cell Signaling, 5335, 1/250 for IF), anti-rabbit IgG, HRP-linked antibody (Cell Signaling, #7074, 1/5000 for WB), anti-rat IgG, HRP-linked antibody (Cell Signaling, #7077, 1/10000 for WB), Alexa Fluor Plus 405 phalloidin (Invitrogen, A30104, 1/200), Alexa Fluor 546 phalloidin (Thermo Fisher, A22283, 1/200), DAPI (BD Bioscience, 564907, 1/1000), goat anti-rabbit IgG Secondary Antibody Alexa Fluor Plus 594 (Invitrogen, A32740, 1/200), goat anti-rat IgG Secondary Antibody Alexa Fluor 488 (Invitrogen, A-11006, 1/200), saponin (Sigma, 8047-15-2), purified rat anti-mouse CD16/CD32 (Mouse BD Fc Block) (BD Pharmingen 553142), Triton X-100 (Sigma, CAS# 9036-19-5), Fluoromount-G (Southern Biotech, 0100-01), RIPA Lysis and Extraction Buffer (Thermo Fisher, 89900), protease inhibitor cocktail (Roche, 11697498001), benzonase (Sigma, E1014-5KU), Laemmli sample buffer (Bio-Rad, 1610747), NuPAGETM Sample reducing agent (Invitrogen, NP0004), gels, and materials for gel migration and membrane transfer were purchased from Bio-Rad, Clarity Western ECL Substrate (Bio-Rad, 1705060).

## Drugs and inhibitors

Latrunculin A (abcam, ab144290, incubation $2\,\mu M$ for 1 hr), para-nitroBlebbistatin (Optopharma, 1621326-32-6, incubation $20\,\mu M$ for 1 hr), nocodazole (Sigma, M1404, incubation $5\,\mu M$ for 1 hr), MLSA1 (Tocris, 4746, incubation $1\,\mu M$ for 1 hr), and SAHA (Tocris, 4652, incubation $10\,\mu M$ for 1 hr). For all experiments in microfluidic chips involving drugs, chips were filled with media + drug (or DMSO) at least 1 hr before experiment, and only media + drug was used at each step.

## Experimental protocols

### Droplet stock formulation

#### Oil phase

$150\,\mu L$ of DSPE-PEG(2000) Biotin solution (10 mg/mL in chloroform) in 30 g of soybean oil, left >4 hr in a vacuum chamber to allow chloroform evaporation. *Aqueous phase*: 10 g of 1% sodium alginate, 15% Pluronic F68 solution in deionized water, gently mixed with a spatula to avoid bubbles. The oil phase was slowly added to the aqueous phase, starting by 2–3 drops, gentle stirring until oil was incorporated, then repeating. Over time, the oil phase incorporates more easily and could be added faster, until a white emulsion was obtained. The emulsion was then sheared in a Couette cell (**Mason and Bibette, 1996**) at 150 rpm to obtain droplets of smaller and more homogeneous diameter. The new emulsion was recovered as it got out of the Couette cell and was now composed of $25\%\frac{v}{v}$ aqueous phase containing $15\%\frac{w}{v}$ Pluronic F68. To wash and remove the smallest droplets, the droplet emulsion was put in a separating funnel for 24 hr at 1% Pluronic F68, 5% oil phase. This operation was repeated at least two times. The final emulsion was stored in glass vials at 12°C, and droplets had a median diameter of $9.4\,\mu m$.

This type of droplets was previously characterized using the pendant drop technique (**Ben M'Barek et al., 2015**; **Molino et al., 2016**) and appears like a relatively stiff substrate (surface tension 12 mN·m$^{-1}$ measured by the pendant drop technique (**Powell et al., 2017**), equivalent to a Laplace pressure of 4.8 kPa for a droplet of radius $5\,\mu m$). The antigen concentration is estimated to be of the order of 50 mol/$\mu m^2$ (see **Pinon et al., 2018** for method) and the diffusion constant ~0.7 $\mu m^2 \cdot s^{-1}$, measured by FRAP, comparable to lipid bilayers (**Bourouina et al., 2011**; **Dustin et al., 1996**; **Zhu et al., 2007**; **Sterling et al., 2015**).

### Droplet functionalization

Droplets were functionalized on the day of experiment. All steps were performed in low binding eppendorfs (Axygen Microtubes MaxyClear Snaplock, 0.60 mL, Axygen MCT-060-L-C), and using PB + Tween 20 buffer (Tween 20 at $0.2\%\frac{v}{v}$ in PB Buffer pH = 7, 20 mM). A small volume of droplet emulsion (here $2\,\mu L$) was diluted 100 times in PB + Tween 20 buffer, and washed three times in this buffer. Washes were performed by centrifuging the solution for 30 s at 3000 rpm in a minifuge, waiting 30 s and then removing $170\,\mu L$ of the undernatant using a gel tip, then adding $170\,\mu L$ of PB + Tween 20. At the last wash, a solution of $170\,\mu L$ + $2.5\,\mu L$ of fluorescent streptavidin solution (1 mg/mL) was added to the droplet solution, then left on a rotating wheel for 15 min, protected from light. Droplets were then washed three times, and at the last wash a solution of $170\,\mu L$ PB + Tween 20 + $5.7\,\mu L$ of biotin goat F(ab')$_2$ anti-mouse IgG (1 mg/mL) (or other biotinylated protein in the same proportion) was added and left to incubate for >30 min on a rotating wheel, protected from light. Droplets were finally washed three times before use, with PB + Tween 20. For experiments using drug treatments, droplets were resuspended in culture media + drug before the experiment.

### Microfluidic chip fabrication

Microfluidic chips were made using an original design from the team of Jacques Fattaccioli (ENS Paris, IPGG) (**Mesdjian et al., 2021**). RTV PDMS was mixed at a ratio 1:9, poured in epoxy cast replicates of the microfluidic chips, and cooked until fully polymerized. Microfluidic chips were then cut, and 0.5 mm-diameter holes were made at the entry/exit sites. The PDMS chip and a FluoroDish were then activated in a plasma cleaner (PDC-32G Harrick) for 1 min and bonded to each other for 1 hr at 60°C. Bonded chips were activated in the plasma cleaner for 1 min to be activated and filled using a syringe with a $0.2\%\frac{w}{v}$ PVP K90 solution in MilliQ water to form an hydrophilic coating. Microfluidic chips were then kept at 4°C in the $0.2\%\frac{w}{v}$ PVP K90-filled FluoroDish to prevent drying for up to a week before the

experiment. On the day of the experiment, microfluidic chips were moved gradually to room temperature (RT), then into an incubator, before imaging. For experiments using drug treatments, microfluidic chips were injected with culture media + drug in the morning and left to incubate to ensure stable drug concentration during the experiment.

## Live imaging of IIA1.6 cell polarization in microfluidic chips

Live imaging of polarization was performed using an inverted spinning disk confocal microscope (Eclipse Ti Nikon/Roper spinning head) equipped with a Nikon ×40, NA 1.3, Plan Fluor oil immersion objective, a CMOS BSI Photometrics camera (pixel size 6.5 μm), and controlled with the Metamorph software (Molecular Devices, France). Stacks of 21 images (δz = 0.7 μm) were taken every 30 s during 40 min, with a binning of 2. Auto Focus was implemented in Metamorph using the bright-field image, then applied to fluorescent channels with a z-offset at each time point. On the day of the experiment, droplets were functionalized and cells were resuspended at $1.5×10^6$ cells/mL in CLICK + 25 mM HEPES. Microfluidic chips, cells, and media were kept in an incubator at 37°C with 5% $CO_2$ until imaging. Droplets (diluted 1/6 from functionalized solution) were injected in the microfluidic chip using a Fluigent MFCS-EZ pressure controller, Tygon tubing, and metal injectors from the dispensing needles 23GA. When enough traps contained a droplet, the inlet was changed to CLICK + 25 mM HEPES (or CLICK + 25 mM HEPES + drug) to rinse PB + Tween 20 buffer and remove any antigen in solution or droplet that could remain. After a few minutes, the inlet was changed to the cell suspension, keeping a minimum pressure to avoid cells encountering droplets before acquisition was launched. Stage positions were selected and the acquisition was launched. After one time point (to have an image of droplets without cells and ensure to have the first time of contact), the inlet pressure was increased to inject cells and create doublets. After 2–5 min (when enough doublets had formed), the injection pressure was lowered to a minimum to limit cell arrival and perturbation of cells by strong flows. For primary B cells, cells were used at $3×10^6$ cells/mL in their media and were imaged using a Nikon ×60, NA 1.4, Plan Fluor oil immersion objective. Stacks of 21 images (δz = 0.7 μm) were taken every 45 s, with a binning of 1.

## Multiple synapse experiments and imaging

For multiple synapse experiments of *Figure 8A, B, E and F*, $2.5×10^5$ cells in 25 μL media were mixed with 4 μL of concentrated droplets (droplet solution washed with media from which the undernatant has been removed as much as possible) and left to interact 2 min at 37°C, before adding 400 μL media to limit new encounters between cells and droplets. This suspension was then added on a FluoroDish coated with 100 mg/mL BSA and left at 37°C. After 45 min, cell–droplet pairs were imaged all over the dish using an inverted spinning disk confocal microscope (Eclipse Ti Nikon/Roper spinning head) equipped with a Nikon ×60, NA 1.4, Plan Fluor oil immersion objective, a CMOS BSI Photometrics camera (pixel size 6.5 μm), and controlled with the Metamorph software (Molecular Devices). Stacks of 21 images (δz = 0.7 μm) were taken, with a binning of 2. Most cells interacted with only two droplets, so only those were considered. For each cell, the number of immune synapses (1 if droplets are close to each other and antigen patches are in the same area, 2 if droplets are apart or antigen patches indicate that the cell interact with the droplets in different places) was determined manually. For multiple synapse experiments following F-actin enrichment and droplet movement in time in *Figure 8C and D*, the experiment was performed in the microfluidic chip to facilitate analysis and started as a typical IIA1.6 polarization experiment. After injection of cells and formation of a few cell–droplet doublets, the inlet was changed back to droplets in order to follow in time the interaction of a cell with two droplets and to image actin enrichment at both synapses easily, acquiring images every 1 min, for 20 min.

## Migration experiment

A homemade PDMS chamber (to limit flows and volumes needed) was bonded to a FluoroDish before coating the glass bottom with 100 mg/mL BSA. The chamber was then filled with media (or media + drug), without HEPES. Cells were pre-reated with drugs, and for each sample $2.5×10^5$ cells were put in 25 μL media and mixed with 3 μL of concentrated droplets and left to interact 2 min at 37°C, before adding 400 μL media to limit new encounters between cells and droplets. This suspension was then added to the PDMS chamber, which was covered with media + drug to prevent drying during

Cell Biology | Immunology and Inflammation

time-lapse imaging. After 30–45 min of cell–droplet encounter, cells were imaged every 4 min for 14 hr using an epifluorescence Nikon TiE video-microscope equipped with a cooled CCD camera (HQ2, Photometrics, pixel size 6.45 μm) and controlled with the Metamorph software (Molecular Devices), using a ×20 (NA = 0.75) dry objective and a binning of 2. During this time-lapse, cells were kept at 37°C with 5% $CO_2$ and imaged in bright field, as well as in 562/40 (red) to visualize the droplet.

## Immunofluorescence with droplets

To approach the non-adherent condition of the cells in the microfluidic chips, IIA1.6 cells were seeded for 15 min on glass coverslips (Marienfeld Superior Precision Cover Glasses, 12 mm diameter) coated with 100 µg/mL BSA, on which they display limited spreading. Droplets were prepared as for live imaging, then diluted 13 times in CLICK + HEPES. A small volume of this droplet solution was deposited on parafilm, and the coverslip was then flipped onto the droplets and left for 5 min, so that droplets would float up to encounter the cells. Coverslips were then put in pre-heated CLICK + HEPES media in a 12-well plate, with the cells facing up, for 0–40 min depending on the time point studied. All manipulations and washes were performed very gently using cut pipet tips to limit cell and droplet detachment. Samples were fixed for 12 min at RT using 4% PFA in PBS, then washed three times with PBS. For imaging of actin in siCtrl, siGEF-H1, DMSO vs. nocodazole, or for imaging of GEF-H1 or EXOC7, samples were incubated 30 min with PBS/BSA/saponin 1×/0.2%/0.05%, then 1 hr at RT with primary antibodies in PBS/BSA/saponin 1×/0.2%/0.05%, followed by three washes with PBS and 1 hr at RT with secondary antibodies in PBS/BSA/saponin 1×/0.2%/0.05%. After three washes with PBS, samples were mounted using Fluoromount-G and left at RT until dry. For acetylated tubulin imaging, samples were permeabilized 5 min with Triton 0.1 %, washed with PBS, then blocked with PBS + 0.2% BSA + 1/200 Fc Block for 10 min. Samples were incubated with primary antibodies diluted in PBS + 0.2% BSA for 1 hr, washed three times with PBS, then incubated with secondary antibodies diluted in PBS + 0.2% BSA 1 hr before being washed and mounted using Fluoromount-G.

3D SIM imaging was performed using a Delta Vision OMX v4 microscope, equipped with an Olympus ×100, NA 1.42, Plan Apo N, oil immersion objective, and EMCCD cameras. Image reconstruction was performed using the SoftWoRx image software under Linux. 3D visualization for figures was performed using the Imaris Viewer software.

Laser scanning confocal imaging was performed using a Leica SP8 laser scanning microscope equipped with a ×40 NA 1.3 oil immersion objective.

## Western blot

B cells were lysed for 10 min at 4°C in RIPA Lysis and Extraction Buffer supplemented with protease inhibitor cocktail, then treated with benzonase. Lysates were spinned for 15 min at 4°C at maximum speed to remove debris, followed by heating of supernatants for 5 min at 95°C with Laemmli sample buffer and NuPAGE sample reducing agent. Supernatants were loaded onto gels and transferred to PVDF membranes. Membranes were blocked for 45 min at RT with 5% BSA in TBS + 0.05% Tween 20, incubated overnight at 4°C with primary antibodies, then incubated 1 hr at RT with secondary antibodies. Membranes were revealed using Clarity Western ECL Substrate, and chemiluminescence was detected using a Bio-Rad ChemiDoc MP imaging system. Western blots were quantified using ImageLab.

## Image and statistical analysis

Image analysis was performed on the Fiji software (*Schindelin et al., 2012*) using custom macros, unless stated otherwise. All codes are available upon request. Single kinetic curves analysis were performed using RStudio (*RStudio, 2020*). Graphs and statistical analysis were made using GraphPad PRISM version 9.2.0 for Windows (GraphPad Software, San Diego, CA, https://www.graphpad.com). All replicates are biological replicates, and the number of replicates is indicated in each figure legend. For graphs of polarization in time of BSA vs. αIgG (*Figure 2*), a moving-average filter of length 3 was applied on the mean and SEM before plotting. The non-smoothed mean curve is superimposed to the graphs. For image analysis of live imaging, cell–droplet doublets were cropped from original acquisitions and were cut so that cells arrive at the second frame (marked as 0 s in figures).

## Analysis of antigen recruitment on the droplet

Bleaching of fluorescent streptavidin was corrected before analysis using Bleach Correction–Histogram Matching. Antigen recruitment was measured by computing the ratio between fluorescence intensity at the synapse and fluorescence intensity at the opposite side on three planes passing through the droplet and the cell, normalized by this value at the time of cell arrival (*Figure 1D*).

## Analysis of F-tractin-tdTomato

Fluorescence was corrected using the Bleach Correction–simple ratio program. Using a custom Fiji macro, 3D masks of the droplet and the cell were generated. Enrichment of F-actin at the immune synapse was defined as the sum of intensity in the mask of the cell within a 2 µm layer around the droplet in 3D, divided by the sum of intensity in the mask of the cell. This measurement was normalized by its value at the first time point of encounter between the cell and the droplet to compensate for potential heterogeneity of the initial state. Extraction of characteristic values (time of peak, maximum) was done with R on single kinetic curves smoothed using 3R Tukey smoothing (repeated smoothing until convergence) (*Tukey, 1977*). Time and value of maximum were computed in the first 10 min of cell–droplet contact. Shape characteristics of the cell (aspect ratio, solidity) were measured on maximum z-projections of cell masks.

## Analysis of C1δ-GFP DAG reporter

Fluorescence was corrected using the Bleach Correction–simple ratio program. Using a custom Fiji macro, 3D masks of the droplet were generated. Enrichment of C1δ-GFP (C1 domain of PKCδ, acting as a DAG reporter; *Botelho et al., 2000*) was defined as the sum of intensity within a 1 µm layer around the droplet. This measurement was normalized with its value at the first time point of encounter between the cell and the droplet to account for variability of reporter expression between cells. Extraction of characteristic values (time of peak, maximum, plateau value relative to maximum) was done with R on single kinetic curves smoothed using 3R Tukey smoothing (repeated smoothing until convergence) (*Tukey, 1977*). Time and value of maximum were computed in the first 10 min of cell–droplet contact.

## Analysis of the centrosome

The 3D movie was first interpolated to obtain isotropic voxels for the advanced analysis. Using a custom Fiji macro, 3D mask of the droplet was generated and position of the centrosome (stained with SiRTubulin) was detected to measure the distance of the centrosome from the droplet surface. Characteristic times were extracted on single kinetic curves smoothed using 3R Tukey smoothing (repeated smoothing until convergence) (*Tukey, 1977*) using R and defined as the first time for which the distance is below 2 µm (only for trajectories starting at >3 µm in order to be able to truly detect the polarization process). This threshold value was chosen looking at the distribution of plateau values for BSA- or αIgG-coated droplets. Tracking of the cell for analysis of centrosome orientation was performed by first obtaining a mask of the cell from SiRTubulin background cytoplasmic signal. This channel is used to create a mask of the cell on Fiji and find its center of mass. Briefly, the 3D stack is interpolated (to obtain an isotropic voxel) and a background subtraction (based on a Gaussian filtered [radius = 4] image of the field without cell, time = 0) is applied. A Gaussian filter is applied on the resulting image (radius = 2) to remove local noise and the cell is finally segmented using an automatic threshold (Huang). Advanced analysis of centrosome trajectories was performed by using the 3D cell contour generated on Fiji, and then computing the distance of the centrosome from the center of the cell, and the angle formed with the cell–droplet axis on MATLAB, to merge this data with advanced nucleus analysis data. For experiments using nocodazole, the centrosome was visualized by expressing eGFP-cent1 and tracked in the same way.

## Analysis of the Golgi apparatus

This was performed on Icy Bioimage analysis software (*de Chaumont et al., 2012*). 3D masks of the Golgi apparatus and the droplet were obtained, and the average distance of the Golgi apparatus to the surface of the droplets was computed using a 3D distance map from the droplet. Characteristic times were extracted on single kinetic curves smoothed using 3R Tukey smoothing (repeated

smoothing until convergence) (*Tukey, 1977*) using R and defined as the first time for which the distance is below 4 µm (only for trajectories starting at >5 µm in order to be able to truly detect the polarization process). This threshold value was chosen looking at the distribution of plateau values for BSA- or αIgG-coated droplets.

## Analysis of the lysosomes

This was performed using Icy Bioimage analysis software (*de Chaumont et al., 2012*). 3D masks of the lysosomes and the droplet were obtained, and the average distance of all the lysosomes to the surface of the droplet was computed using a 3D distance map from the droplet. Characteristic times were extracted on single kinetic curves smoothed using 3R Tukey smoothing (repeated smoothing until convergence) (*Tukey, 1977*) using R and defined as the first time for which the distance is below 3 µm (only for trajectories starting at >4 µm in order to be able to truly detect the polarization process). This threshold value was chosen looking at the distribution of plateau values for BSA- or αIgG-coated droplets.

## Analysis of the nucleus and detection of nuclear indentation

This was performed using custom Fiji macros and MATLAB software (available upon request). B cell nucleus is bean-shaped and exhibits a marked invagination. To automatically detect the invagination at each time point, we interpolated the confocal images of the nucleus to obtain an isotropic voxel, segmented the nucleus, and found the interpolating surface (isosurface function in MATLAB). We smoothed the surface to reduce voxelization and computed the mean curvature at each vertex with standard differential geometry methods. We defined the invagination as the point with the minimal mean curvature obtained on this surface. Ad hoc correction based on nearest-neighbor tracking is applied when several local minima are found (in nuclear that exhibit several lobes), the selected minimum is the nearest one to the point found in the previous frame. The orientation of the nucleus with respect to the $Cell_{Center}$–$Droplet_{Center}$ axis is quantified as the angle $N_{indentation}$–$Cell_{Center}$–$Droplet_{Center}$.

## Analysis of actin profiles in OMX images

This was performed using custom Fiji macro. Mask based on droplet fluorescence is built and fitted to a 3D ellipsoid and the voxels made isotropic (bilinear interpolation). The ellipsoid box is centered and 3D rotated so that the axis of the ellipsoid is oriented along the reference frame (the largest corresponding to the x-axis and the shortest to the z-axis). The same roto-translations are applied to the actin channels to orient it on the x–y plane. Line scans are symmetric radial scan obtained from an average projection of 25 planes (i.e. 1 µm) centered on the ellipsoid center. Graph are plotted after normalization to the maxima.

## Analysis of immunofluorescence of GEF-H1 and EXOC7

This was performed using custom Fiji macros. One plane in the center of the synapse was used for GEF-H1, and six planes (δz = 0.34 µm) centered around the immune synapse were used for EXOC7. Masks of the droplet and the cell were obtained. Enrichment at the immune synapse was measured as the ratio between the integrated fluorescence intensity of the staining (GEF-H1 or EXOC7) within 1 µm of the droplet, in the cell mask, and the total integrated fluorescence intensity.

## Analysis of immunofluorescence of F-actin polarized distribution

This was performed using custom Fiji macros. F-actin intensity was measured over six planes around the immune synapse (δz = 0.34 µm), doing a linescan spanning the width of the cell, going from the immune synapse to the cell rear. Profiles were then normalized for cell length.

## Analysis of immunofluorescence of acetylated tubulin

This was performed using custom Fiji macros. 3D masks were obtained using the phalloidin staining, and the integrated fluorescence intensity in the mask was computed for α-tubulin and acetylated α-tubulin.

## Analysis of cell migration experiments

This was performed using manual tracking in TrackMate (*Tinevez et al., 2017*), tracking only cells in contact with one droplet, and stopping the track before cell division when this occurred. Trajectories were then analyzed on R using the trajr package (*McLean et al., 2018*). To compute the confinement ratio and the mean directional change rate, only trajectories of migrating cells (distance between initial and final position >20 μm) were considered, starting the trajectory at the beginning of migration (distance between two consecutive images >6 μm, the radius of the cell), and for the 30 following frames, corresponding to a 2 hr movie.

## Data availability

All data generated or analyzed during this study are included in the source data files and supporting files. Custom image analysis scripts are available online at https://github.com/PierobonLab/Paper-Pineau2022, (*Pineau, 2022*, copy archived at swh:1:rev:85d6b03e630656509de21146c6b199258de70659). The following source codes from the GitHub repository were used to analyze the images:

Antigen_recruitment Fiji macros to quantify antigen recruitment. Masks can be generated from the fluorescent or the transmission channel (less resolved).

ActinLive_Analysis Fiji macros to obtain masks of the cell and the droplet, count the number of actin maxima and their distance to the immune synapse, cell shape characteristics and measure the actin enrichment within 2 μm of the immune synapse. Cell shape analysis code was also used to quantify nuclear shape.

Cell_Nuc_Mtoc Fiji macros to segment droplet, nucleus, cells, and MTOC, and find the distances of the organelles from the droplet, and the orientation of the centrosome.

Synapse_Linescan Fiji macros to analyze actin profile at the synapse from 3D images (possibly OMX 3D SIM).

DAGReporter_Analysis Fiji macros to obtain masks of the cell and the droplet and measure the enrichment of DAG reporter within 1 μm of the immune synapse.

Lyso_Drop Icy Bioimage analysis protocol to measure the lysosome–droplet distance.

Golgi_Drop Icy Bioimage analysis protocol to measure the Golgi apparatus–droplet distance.

GEFH1_Analysis Fiji macros to quantify enrichment of GEF-H1 at the immune synapse on one plane, within 1 μm of the droplet, on immunofluorescence images.

EXOC7_Analysis Fiji macros to quantify enrichment of EXOC7 at the immune synapse on six planes, within 1 μm of the droplet, on immunofluorescence images.

AcetylTub_Analysis Fiji macros to generate a mask of the cell and the droplet from IF of micro-tubules and compute the ratio between acetylated and total α-tubulin.

ActinPolarityLinescan_Analysis Fiji macros to generate a mask of the cell and the droplet on immunofluorescence images, and do a linescan of F-actin intensity along the cell polarity axis on six planes.

Nuclear_Shape Fiji macro to prepare the image to be analyzed with the MATLAB codes (see Readme.txt) to obtain the orientation of the nucleus based on the position of its indentation.

## Acknowledgements

The authors thank H Moreau and A Yatim for their help in setting up the migration experiment, M Bolger-Munro for critical reading, H Moreau, J Delon, M Théry, J Husson, and C Hivroz for fruitful discussions, and acknowledge the Nikon Imaging Center and CNRS-Institut Curie, the PICT-IBiSA, Institut Curie, Paris, member of the France-BioImaging national research infrastructure, and the Cell and Tissue Imaging Platform-PICT-IBiSA (member of France–Bioimaging ANR-10-INBS-04) of the Genetics and Developmental Biology Department (UMR3215/U934) of Institut Curie (supported by the European Research Council [ERC EPIGENETIX N°250367]) for support in image acquisition, as well as the Recombinant Antibody Platform (TAb-IP) of Institut Curie. This work benefited from the technical contribution of the Institut Pierre-Gilles de Gennes joint service unit CNRS UAR 3750. The authors would like to thank the engineers of this unit for their advice during the development of the microfluidic setup. PP and AMLD were supported by CNRS and INSERM, respectively. JP was funded by the Ecole Doctorale FIRE-Programme Bettencourt, Université de Paris and by funding from the Agence Nationale de la Recherche (ANR-21-CE30-0062-01 IMPerIS) and LP was funded by the IPV

scholarship program (Sorbonne Université). PP, AMLD, and JP acknowledge funding from the Agence Nationale de la Recherche (ANR-10-IDEX-0001-02 PSL*) and JF acknowledges funding from the Agence Nationale de la Recherche (ANR Jeune Chercheur PHAGODROP, ANR-15-CE18-0014-01).

## Additional information

### Funding

| Funder | Grant reference number | Author |
|---|---|---|
| Agence Nationale de la Recherche | ANR-15- CE18-0014-01 | Jacques Fattaccioli |
| Agence Nationale de la Recherche | ANR-21-CE30-0062-01 IMPerIS | Judith Pineau |
| Agence Nationale de la Recherche | ANR-10-IDEX-0001-02 PSL* | Judith Pineau |

The funders had no role in study design, data collection and interpretation, or the decision to submit the work for publication.

### Author contributions
Judith Pineau, Conceptualization, Resources, Data curation, Software, Formal analysis, Funding acquisition, Investigation, Visualization, Methodology, Writing – original draft, Writing – review and editing; Léa Pinon, Resources, Methodology; Olivier Mesdjian, Resources, Validation, Methodology; Jacques Fattaccioli, Funding acquisition, Validation, Methodology; Ana-Maria Lennon Duménil, Conceptualization, Supervision, Funding acquisition, Writing – original draft, Project administration, Writing – review and editing; Paolo Pierobon, Conceptualization, Data curation, Software, Formal analysis, Supervision, Validation, Investigation, Writing – original draft, Project administration, Writing – review and editing

### Author ORCIDs
Judith Pineau http://orcid.org/0000-0003-0665-1210
Paolo Pierobon http://orcid.org/0000-0002-3014-0181

### Decision letter and Author response
Decision letter https://doi.org/10.7554/eLife.78330.sa1
Author response https://doi.org/10.7554/eLife.78330.sa2

## Additional files

### Supplementary files
• MDAR checklist

### Data availability
All data generated or analyzed during this study are included in the manuscript source data files and supporting files. Custom image analysis scripts are available online at https://github.com/PierobonLab/Paper-Pineau2022, (copy archived at swh:1:rev:85d6b03e630656509de21146c6b199258de70659).

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
