## [Editor Report]

This study provides new insights into the fundamental process of immune cell synapse formation in the context of B-lymphocyte antigen receptors and cognate antigen-presenting cells or surfaces. The authors use elegantly designed microfluidic systems, allowing control of the nature and number of the antigenic surfaces presented, and at the same time an unprecedented view of the process of immune synapse maturation in situ. These experiments provide an understanding of the specific roles of the reorganization of the actin as well as the microtubule cytoskeleton in the selection and restriction of a unique immune synapse, an important process in high-affinity antibody generation.

---

## [Decision Letter]

[Editors' note: this paper was reviewed by Review Commons.]

---

## [Author Response]

1. General Statements

In this work, we investigated the cytoskeleton rearrangement events leading to synapse formation in the context of antigen recognition by B cells. We developed a microfluidics-based device to quantify the dynamics of these events in space and time and to study their coordination. We observed two types of events: early local events, which take place at the synapse and are controlled by the actin cytoskeleton, and late global events that involve the polarization of additional organelles and require microtubules. Noticeably, we found that these two phases are not independent as microtubules are needed to restrict actin polymerization to the immune synapse via the GEF-H1-RhoA signaling pathway. This allows B cells to form and maintain over time a single polarity axis, promoting the formation of a singular polarity axis.

All three reviewers have understood the work and raised competent and interesting critiques. We appreciate that they saw the paper’s significance for both the immunology and cell biology communities. Referees #2 and #3 are very positive on the results and mainly raise technical control concerns that we will address. Referee #1 believes that the results presented in figures 6 & 8 deserve to be strengthened. Therefore, our experimental plan aims at strengthening these results by providing additional evidence for the role of microtubules at the B cell immune synapse in promoting a single polarity axis by restricting actin polymerization to the synapse via the GEFH1-RhoA pathway.

2. Description of the planned revisionsReferee #1:1) The data presented in support of the concluding model (that microtubules enforce synapse uniqueness by constraining the scope of GEF-H1) fall short of being convincing. Although GEFH1 depletion appears to rescue the cell shape defects induced by nocodazole (Figure 6B), there is no quantitative analysis of the effects of GEF-H1 depletion on the depolarization of F-actin. All we are shown is a single high resolution micrograph (Figure 6E). Furthermore, the capacity of GEF-H1 depletion to rescue the multiple synapse phenotypes explored in Figure 8 is not assessed. I appreciate that depletion of GEF-H1 could diminish F-actin accumulation at the synapse, but it should be possible to employ an alternative signaling readout for synapse formation. Alternatively, the authors could hyperactivate the GEF-H1 pathway using perturbations of microtubule acetylation or a GEF-H1 mutant that does not bind to microtubules. In general, more data is required that specifically implicates GEF-H1 in this process.

In Figure 6B, we show GEF-H1 accumulation at the immune synapse with time, which is impaired upon microtubule depolymerization. We also established the impact of microtubule acetylation (increased using SAHA, a histone deacetylase inhibitor) on F-actin and GEF-H1 relative enrichment at the immune synapse in Figure 6C-D.

We quantified the distribution of F-actin along the cell-droplet axis and found that siGEFH1 prevents the effect of Nocodazole treatment on its distribution (Figure 6-Supplement Figure 1D-E).

We included new data on GEF-H1 localization and actin localization in GEF-H1 silenced cells in Figures 6B-C-D and 6I. Actin distribution along the polarity axis is also shown in Figure 6-Supplement Figure 1D-E, and additional examples are added in Figure 6-Supplement Figure 2.

We performed new experiments mixing cells and droplets outside the chip and considered only cells in contact with two droplets (having exclusively single or double synapses as a possible outcome). The new data include GEF-H1-silenced cells, and are reported in Figures 8 A, B, E. In addition, we showed that the presence of multiple synapses can also be promoted by increased microtubule acetylation using SAHA (histone deacetylase inhibitor), see Figure 8F.

We included measurements of antigen recruitment at the immune synapse in control of GEF-H1-silenced cells, as shown in Figure 6-Supplement Figure 1C.

We used SAHA, a histone deacetylase inhibitor, to increase microtubule acetylation. Indeed, Tubacin treatment induced a strong destabilization of our oil droplets, likely due to its chemical structure. SAHA treatment led to a decreased relative enrichment in GEF-H1 at the immune synapse (Figure 6C), a decrease in the polarization of F-actin (Figure 6D), and the promotion of multiple synapses (Figure 8F). These effects go in the same direction as microtubule disruption.

2) The phenotypes in Figure 8 are surprisingly subtle, particularly given how strong the effect of nocodazole is on F-actin polarization in single B cell-droplet conjugates (Figure 5D, H). One would expect an even stronger F-actin depolarization phenotype when an actual second target is present (e.g. Figure 8B). Instead, panel C indicates a very weak phenotype (perhaps not even statistically significant?). Differences are also quite small in Figure 8D. The subtlety of these phenotypes, despite the fact that a rather dramatic perturbation was applied (microtubule depolymerization) leads me to wonder whether microtubules are only peripherally involved in maintaining synapse uniqueness, and/or whether there might be a compensatory pathway. Perhaps the microtubule-GEF-H1 axis may be more important for restricting migration away from the synapse rather than the formation of multiple synapses? This alternative interpretation may be more consistent with the Figure 5 and Figure 8 data.

The new quantification and synapse counting method improved consistently the significance of the difference between untreated and Nocodazole-treated cells.

We performed a series of cell motility experiments with activated B cells (on BSA-treated surfaces as the confined assay turned out to be less controllable). We showed that the trajectories of nocodazole-treated cells are less persistent than the ones of untreated cells, see Figures 8G-I, consistent with the idea that microtubules disruption destabilizes the single polarity axis.

3) Figures 5L and 6E should be quantified over multiple synapses.

This new quantification is now presented in Figures 5L and 6I.

4) Key results should be reproduced in primary B cells, preferably using actual antigen-bearing target cells.

As anticipated, we could not present a comprehensive study on primary cells. Nonetheless, we show that our microfluidics system can be adapted to primary murine B cells to observe antigen and actin accumulation at the immune synapse (Figure 1-Supplement Figure 1B-E). Other observations would require setting up transgenic mice models (transfection and Sir Tubulin do not work properly in these cells). We cannot, therefore, reproduce all the data of the paper in a reasonable amount of time.

5) What are the effects of taxol? Do Taxol treated cells form synapses? Can they form second synapses? Is F-actin polymerization at the synapse inhibited?

Taxol-treated cells show extremely variable effects: Antigen recruitment is lower, while actin seems recruited more strongly at the synapse, but with high cell-to-cell variability. Taxol affects several aspects of the cytoskeleton (transport, microtubule state, etc) making these data hard to interpret. Therefore, we present the data here below, but we will not include them in the manuscript (Author response image 1).

**Author response image 1. sa2fig1:** The effect of Taxol (1 µM) on B lymphocyte immune synapse formation. (A) Plateau (Average 25-30min) of antigen recruitment (Median+/- IQR,DMSO N=8;8, Taxol N=9;6, 2 independent experiments, Mann-Whitney test). (B) Timelapse images of F-tractin td Tomato-expressing cells in contact with an antigen-coated droplet, treated with DMSO or Taxol. Scale bar 5µm. (C) F-actin enrichment at the immune synapse (as defined in the paper) in time (Mean+/- SEM). (D) peak value of F-actin enrichment in 0-10min and (E) time of peak of F-actin enrichment (Median+/- IQR, DMSO N=12;11, Taxol N=6;7, 2 independent experiments, Mann-Whitney test).

Referee #21) The manuscript presents very interesting findings. The use of the microfluidics system in combination with the oil droplets as a model antigen-presenting surface is a recent technological advance from this lab (https://www.biorxiv.org/content/10.1101/2021.12.22.473360v2.full.pdf), which I think will bring new understanding to B cell activation. The experiments are hypothesis driven and done well, and the data are presented clearly both with representative images and presentation of analysed data compiled from independent experiments. I do not think that additional experiments are essential to support the claims of the paper. However, I was surprised that the authors did not measure of antigen internalisation from the droplet. The authors discuss the importance of synapse formation for antigen extraction and have a system to monitor both actin polymerisation and lysosome recruitment to the synapse; is it possible to monitor antigen internalisation in these experiments? It would be a very nice addition to the manuscript, though it is not essential to support the main claims.

Illustration and quantification of EXOC7 recruitment at the synapse are shown in Figure 1—Supplement Figure 1A.

2) The figure legends indicate that most experiments were repeated twice, and the bar plots compile all data points (individual cells) from both experiments and the statistical analysis performed on these. There is often a large spread in the data, and it is difficult to know whether this is from cell-to-cell variability in general or variability between the independent experiments. It would be helpful if the authors could color-code the data points by experiment and include the mean value for each independent experiment on the same bar plot. I don't think this will change the conclusions of the paper, but I do think it would clarify how much variability there is in the cell behavior and how reproducible the experiments are.

This has been done.

Minor comments:1) Figure 2G quantifies the characteristic times of polarization events from the imaging measurements of B cell synapse formation. While the mean values suggest a series of events that begin with actin enrichment and DAG signalling, and finish with nucleus rearward transport, there is a very large spread in the data and I wonder whether this representation of the data reflects what happens at the single-cell level. Is it possible to show the timescales of these events at the single cell level?

See Figure 2-Supplement Figure 1.

2) Figure 5K shows a striking oscillation in DAG enrichment in nocodazole-treated cells. It is very interesting that the cells all seem to oscillate with the same frequency even across different experiments. Could the authors comment on this observation?

We believe that, unfortunately, this results from the effect of a single cell on the average value and is therefore not significant (Author response image 2, violet curve).

Referee #3Major concerns1) The role of GEF-H1 should be verified with rescue experiments.

We re-expressed GEF-H1 in GEF-H1-silenced cells and recovered indeed the unperturbed phenotype, see Figure 6-figure supplement 1 A-B.

2) The effect of RhoA on synapse formation and polarization should be investigated more thoroughly. For instance, the effects of RhoA-WT and RhoA-DN constructs should also be shown.

We have expressed RhoA-DN and observed shape changes in unperturbed vs Nocodazole-treated cells (Figure 7—figure supplement 1). Unfortunately, actin distribution in RhoA-DN was not easily quantifiable.

3) Description of the revisions that have already been incorporated in the transferred manuscriptPlease insert a point-by-point reply describing the revisions that were already carried out and included in the transferred manuscript. If no revisions have been carried out yet, please leave this section empty.

We included in the new version of the manuscript:

1. More details in the legend of Figure 8D answering to point 3 referee #1: explain the y axis in Figure 8D more thoroughly

2. Notes answering to point 4 referee #1: in the paragraph referring to Figure 2, we compare DAG kinetics with previous measurements in T cell (Gawden-Bone 2018)

3. More information in the legends (especially with regards to imaging), as asked by the referee #3 (minor point)

4. A paragraph in the discussion where we compare the timeline and timescale measured in our work to the values found in the literature (even if in other antigen presentation systems or other cell types), as asked by referee #3 (minor points)

5. The word “quantification” in Figure 6A has been removed (as suggested by referee #3)

4) Description of analyses that authors prefer not to carry outPlease include a point-by-point response explaining why some of the requested data or additional analyses might not be necessary or cannot be provided within the scope of a revision. This can be due to time or resource limitations or in case of disagreement about the necessity of such additional data given the scope of the study. Please leave empty if not applicable.Referee #3: Also, siRNA studies should include at least two silencing oligos leading to the same phenotype and a scrambled control. siRNAs can lead to lots of off-target effects and careful validation is thus important.

In our experience siRNA pools work better and are not less specific than single siRNA, even though this might depend on the nature of the target mRNA. In the case of GEF-H1, siRNA Smart Pool systems have been successfully used in other publications (Ho et al., Cell Reports, 2021; Dhakal et al., 2008, Journal of Biological Chemistry; Ridgway 2010, Journal of Cellular Biochemistry). May also the referee keep in mind that the results obtained with GEF-H1 silencing were confirmed by transfection of constitutively active RhoA. In addition, if rescue experiments are successful, this will show that the siRNA pool used is specific.

Rescue experiments were successful: we consider that the phenotype is specifically due to GEF-H1 silencing and we added a comment in the text.

Referee #3: Also a very interesting approach here, although probably beyond the scope of this study, could be to use, for example, photoinducible constructs to focally activate/inactive the cytoskeletal regulation.

An optogenetic approach, though very elegant, is technically very challenging in our experimental system and would therefore require a lot of time to be set up. This would thus considerably delay the publication of our manuscript without significantly changing its message.

Referee #3, as minor concern, asked: Figure 2E: Have the authors tried to measure lysosome motility with any other marker than Lysotracker? Is it possible that the less acidic lysosomes that appear fainter with Lysotracker are approaching the synapse earlier than the more acidic ones (more intense lysotracker)? Can the authors comment about possible vesicle acidity changes during cell activation?

The question of whether lysosomes modify their pH while being recruited to the synapse is indeed very interesting. However, addressing it properly would require a lot of experimental development and time as currently, our results do not have enough spatial and temporal resolution to do so. We believe that taking this time is not reasonable as this point falls out of the scope of our manuscript.